# Septal cholinergic input to CA2 hippocampal region controls social novelty discrimination via nicotinic receptor-mediated disinhibition

Domenico Pimpinella[1], Valentina Mastrorilli[2], Corinna Giorgi[1,3], Silke Coemans[1], Salvatore Lecca[4], Arnaud L Lalive[4], Hannah Ostermann[5], Elke C Fuchs[5], Hannah Monyer[5], Andrea Mele[2], Enrico Cherubini[1], Marilena Griguoli[1,6]*

[1]European Brain Research Institute (EBRI), Fondazione Rita Levi-Montalcini, Rome, Italy; [2]Department of Biology and Biotechnology 'C. Darwin', Center for Research in Neurobiology 'D. Bovet', Sapienza University of Rome, Rome, Italy; [3]Institute of Molecular Biology and Pathology of the National Council of Research (IBPM-CNR), Roma, Italy; [4]Department of Fundamental Neurosciences, University of Lausanne, Lausanne, Switzerland; [5]Department of Clinical Neurobiology of the Medical Faculty of Heidelberg University and German Cancer Research Center (DKFZ), Heidelberg, Germany; [6]Institute of Neuroscience of the National Research Council (IN-CNR), Pisa, Italy

*For correspondence:
m.griguoli@ebri.it

Competing interest: The authors declare that no competing interests exist.

**Abstract** Acetylcholine (ACh), released in the hippocampus from fibers originating in the medial septum/diagonal band of Broca (MSDB) complex, is crucial for learning and memory. The CA2 region of the hippocampus has received increasing attention in the context of social memory. However, the contribution of ACh to this process remains unclear. Here, we show that in mice, ACh controls social memory. Specifically, MSDB cholinergic neurons inhibition impairs social novelty discrimination, meaning the propensity of a mouse to interact with a novel rather than a familiar conspecific. This effect is mimicked by a selective antagonist of nicotinic AChRs delivered in CA2. Ex vivo recordings from hippocampal slices provide insight into the underlying mechanism, as activation of nAChRs by nicotine increases the excitatory drive to CA2 principal cells via disinhibition. In line with this observation, optogenetic activation of cholinergic neurons in MSDB increases the firing of CA2 principal cells in vivo. These results point to nAChRs as essential players in social novelty discrimination by controlling inhibition in the CA2 region.

## Introduction

Acting in the brain as a neurotransmitter and a neuromodulator, acetylcholine (ACh) controls neuronal circuits involved in attention, learning, and memory (for review see *Hasselmo, 2006*). Dysfunction of the cholinergic system contributes to cognitive impairments associated with several neuropsychiatric diseases including neurodevelopmental and neurodegenerative disorders (*Deutsch et al., 2016*; *Hampel et al., 2018*; *Perez-Lloret and Barrantes, 2016*). The medial septum/diagonal band of Broca complex (MSDB) in the basal forebrain provides the major cholinergic innervation to the hippocampus through the fimbria–fornix pathway (for review see *Dutar et al., 1995*). ACh, released from cholinergic terminals via both wired and volume transmission, targets nicotinic and muscarinic receptors (nAChRs and mAChRs) differently distributed in subcellular domains and cell types across the hippocampal layers (*Umbriaco et al., 1995*; *Sarter et al.,*

2009). Activation of nAChRs and mAChRs, which relies on local ACh concentration controlled by the hydrolytic action of acetylcholinesterase (*Vijayan, 1979*), leads to complex effects on neuronal excitability, synaptic plasticity, rhythmic oscillations, brain states, and behavior (for review see *Teles-Grilo Ruivo and Mellor, 2013*; *Dannenberg et al., 2017*; *Haam and Yakel, 2017*).

Recently, the dorsal CA2 and the ventral CA1 hippocampal regions have been associated with social memory, which in rodents can be assessed as the capacity of the animal to discriminate between novel and familiar individuals (*Stevenson and Caldwell, 2014*; *Hitti and Siegelbaum, 2014*; *Okuyama et al., 2016*; *Raam et al., 2017*; for review see *Dudek et al., 2016*; *Piskorowski and Chevaleyre, 2018*). Like other hippocampal subfields, the CA2 receives substantial cholinergic innervation from the MSDB (*Cui et al., 2013*; *Hitti and Siegelbaum, 2014*; *Kohara et al., 2014*); however, the potential role of ACh in social memory has not been explored yet. Here, we report that ChAT+ neurons in the MSDB are activated in response to social stimuli, and that selective inhibition of cholinergic neurons in the MSDB or cholinergic fibers in the dorsal CA2 hippocampal region impairs social novelty discrimination. This effect is mimicked by local delivery of nAChRs but not mAChRs blockers, suggesting a key role of ACh in controlling social novelty discrimination, via nAChRs.

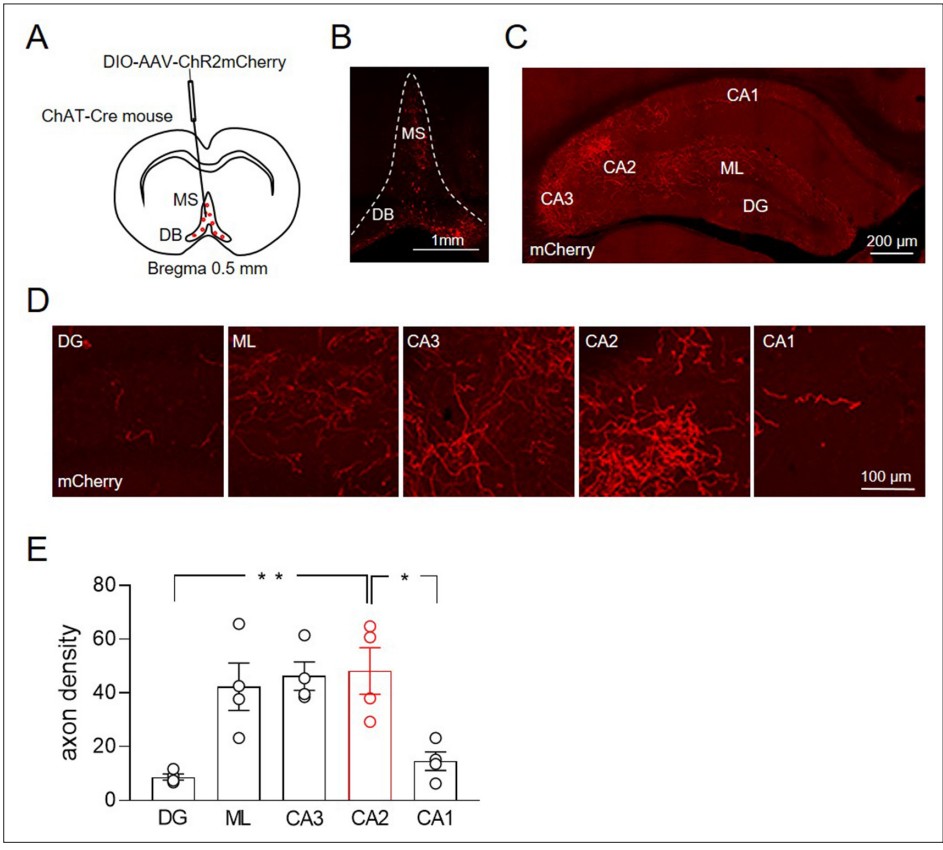

**Figure 1.** Distribution of ChAT+ neurons in the medial septum/diagonal band of Broca (MSDB) and their axon fibers in the hippocampus. (**A**) Schematic drawing showing the injection site of DIO-AAV-ChR2mCherry delivered in the MSDB of ChAT-Cre mice. (**B**) Confocal fluorescent image showing ChR2-mCherry expression in ChAT+ neurons in the MSDB. (**C**) Confocal fluorescent image showing the distribution of ChAT+ axon fibers, which express ChR2-mCherry and innervate different hippocampal regions. (**D**) High magnification images showing ChAT+ axon fibers in the DG, molecular layer (ML), CA3, CA2, and CA1 regions. (**E**) Quantification of septal cholinergic axon densities in the DG, ML, CA3, CA2, and CA1 regions (n = 4 animals; four hippocampal slices/animal; DG: 8.73 ± 1.1; ML: 42.3 ± 8.8; CA3: 46.3 ± 5.3; CA2: 48.2 ± 8.6; CA1: 14.6 ± 3.5; p = 0.0006; one-way ANOVA). Open circles are values from single animals and bars are mean ± SEM. *: p < 0.05; **: p < 0.01.

# Results

## ACh released from cholinergic neurons of the MSDB is required for social novelty discrimination

To evaluate the extent of MSDB cholinergic fibers targeting the hippocampus, we injected an AAV-DIO-ChR2-mCherry virus into the MSDB of 1-month-old male mice-expressing Cre recombinase in acetylcholine transferase-positive neurons (ChAT-Cre) (*Figure 1A*). mCherry was expressed in cells bodies of neurons in the MSDB (*Figure 1B*) and in cholinergic fibers targeting different hippocampal areas (*Figure 1C*). Macroscopic inspection and quantitative analysis revealed a high cholinergic fiber density in molecular layer (ML), CA2 and CA3 regions as compared to CA1 and DG (*Figure 1D-E*).

To validate the hypothesis that MSDB ChAT$^+$ neurons are activated during social tasks, the expression of the activity-dependent gene c-Fos was assessed by immunofluorescence analysis. Counting of MSDB c-Fos$^+$ nuclei was evaluated in home-caged control (HCC) mice or in animals exposed to social interaction (SI) by performing sociability and social novelty discrimination tasks in the three-chamber test (*Moy et al., 2004*). Minimal c-Fos$^+$ labeling was detected in MSDB cells of HCC animals while a strong increase in c-Fos$^+$ nuclei, including a subset of ChAT$^+$ neurons, was observed 1 h after SI (*Figure 2A-C*). To understand whether this was triggered by the environment, the expression of c-Fos was measured in MSDB neurons of animals exposed to the empty arena (EA). In this condition, there was also a significant increase in the percentage of MSDB c-Fos$^+$ nuclei compared to HCC (*Figure 2D-F*). However, the stimulation of c-Fos expression induced by the exposure to EA was significantly less than that observed in response to social stimuli (*Figure 2G*). Furthermore and most interestingly, in the EA condition, none of the c-Fos$^+$ cells were ChAT$^+$ indicating a selective activation of MSDB cholinergic neurons in response to social stimuli (*Figure 2H*).

To assess whether endogenous ACh released from MSDB nuclei is involved in the animal's ability to discriminate between novel and familiar individuals, multiple strategies were used. Firstly, we sought to inactivate ChAT$^+$ neurons in the MSDB. To this end, the MSDB of ChAT-Cre mice was injected stereotactically with an AAV carrying a floxed eYFP, as control, or the Tetanus toxin light chain fused to GFP (TeNT). TeNT contains a zinc-endopeptidase domain that selectively cleaves VAMP/synaptobrevin on synaptic vesicles, hence blocking the neuroexocytosis process (*Schiavo et al., 1992*; *Rossetto et al., 1995*). This approach allows to block the release of ACh from cholinergic terminals. TeNT was efficiently expressed by ChAT$^+$ neurons in the MSDB (*Figure 3—figure supplement 1A*). Patch clamp recordings from MSDB ChAT$^+$ neurons in acute slices allowed to investigate the effect of TeNT expression on intrinsic membrane properties (*Figure 3—figure supplement 1B-D*). As shown in the figure, all ChAT$^+$ neurons were spontaneously active. No significant differences in the resting membrane potential ($V_m$) and in the spontaneous firing frequency between eYFP- and TeNT-expressing neurons were observed (*Figure 3—figure supplement 1B-C*). In addition, as already reported in cultured neurons (*Dimpfel, 1979*), TeNT-expressing neurons exhibited a significant reduction in their spike half-width and in their input resistance ($R_{in}$) (*Figure 3—figure supplement 1D*).

Mice-expressing eYFP or TeNT were subjected to the three-chamber test for both sociability and social novelty evaluation (*Figure 3A-D*). During the sociability task, TeNT-expressing mice were comparable to eYFP mice and showed a preference for the animal over the object (*Figure 3A-B*). To evaluate social novelty discrimination, after 1 h the mouse was subjected to a social novelty task consisting in the exposition to a previously encountered familiar mouse or a novel one. TeNT mice did not show the typical preference for the novel subject compared to the familiar one, as was the case in the control group (*Figure 3C-D*), indicating that the release of ACh is required for social novelty discrimination.

To investigate whether this effect was selective for social recognition, TeNT and control mice were subjected to a general learning and memory test, named novel object recognition (NOR). During NOR, the test animal was first exposed to two identical objects. After 1 h, one object was replaced by a novel one (*Figure 3E*). TeNT mice did not show any impairment in the preference for the novel object as compared to eYFP group (*Figure 3F*). Thus, the NOR task is not dependent on MSDB ChAT$^+$ neurons.

To assess whether the TeNT-induced impairment in social novelty discrimination was associated with deficits in motor behavior and anxiety, mice carrying eYFP or TeNT were tested in the open-field paradigm (*Figure 3—figure supplement 2A*). No differences in distance traveled, speed, or time spent in the outer or inner zone were observed between eYFP and TeNT mice (*Figure 3—figure supplement 2B-D*). Moreover, no difference in distance traveled or speed was found between the two

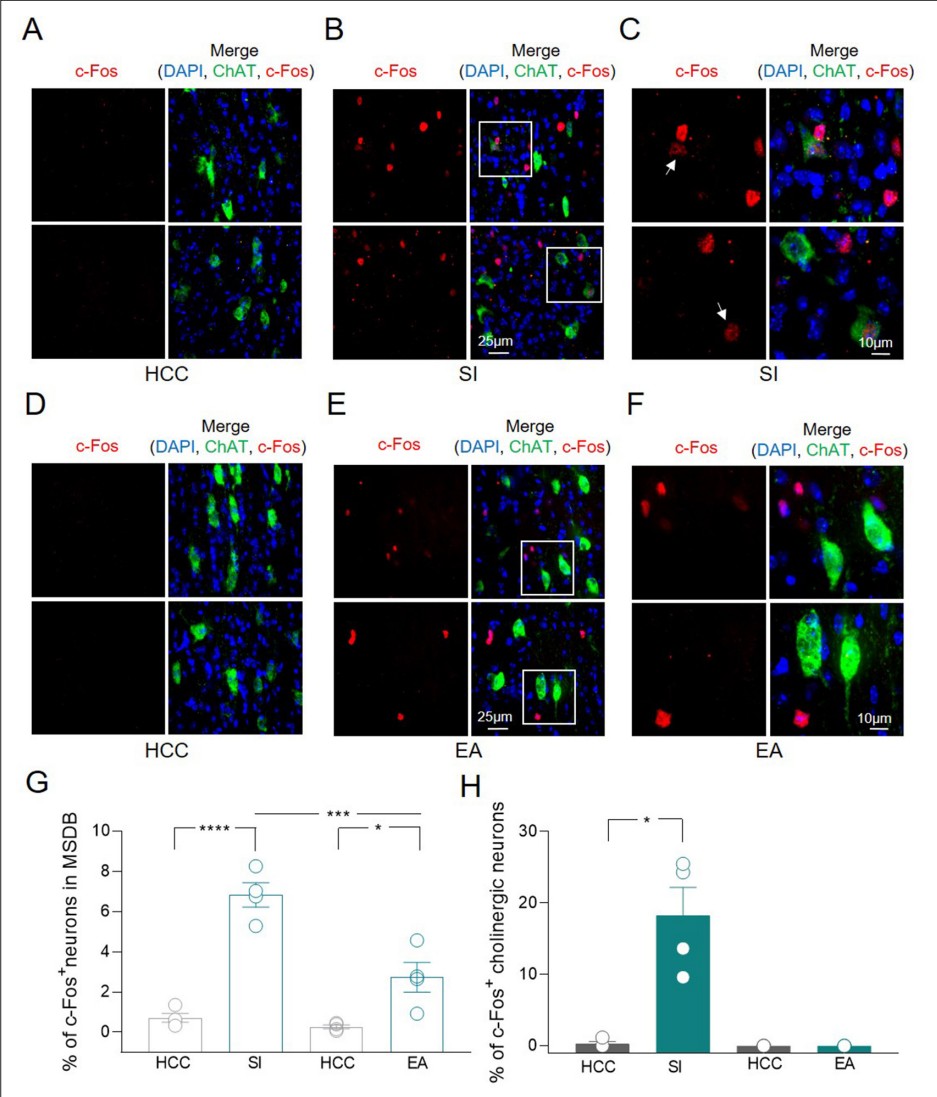

**Figure 2.** c-Fos detection in a subset of medial septum/diagonal band of Broca (MSDB) ChAT[+] neurons following social behavior and environment exposure. (**A–F**) Confocal images of MSDB coronal slices, immunolabeled for ChAT (green) and c-Fos (red) to detect behavior-dependent activation of cholinergic neurons. (**A**) No c-Fos[+] neurons were detected in the MSDB of home-caged controls (HCCs). (**B**) c-Fos labeling of the MSDB from animals subjected to the three-chamber test (social interaction, SI) and sacrificed 1 h later, revealed sparse activation of cells including ChAT[+] neurons. (**C**) High magnification of insets in (**B**) showing representative c-Fos[+] nuclei of cholinergic neurons (white arrows). (**D**) No c-Fos[+] neurons were detected in the MSDB of HCCs. (**E**) c-Fos labeling of the MSDB from animals subjected to three-chamber exploration (empty arena, EA) and sacrificed 1 h later, revealed sparse activation of cells not including ChAT[+] neurons. (**F**) High magnification of insets in (**E**) showing representative c-Fos[+] nuclei of MSDN ChAT[−] neurons. (**G**) Aligned dot plot showing the mean percentage of c-Fos[+] nuclei detected in the MSDB of HCC, SI, and EA experimental groups ($n$ = 4 animals/group; HCC: 0.72% ± 0.2%; SI: 6.83% ± 0.6%, $p < 0.0001$; HCC: 0.26% ± 0.1%; EA: 2.73% ± 0.8%, $p = 0.02$; SI vs EA, $p = 0.0004$, one-way ANOVA). (**H**) Aligned dot plot showing the mean percentage of c-Fos[+] nuclei detected in ChAT[+] neurons of HCC, SI and EA experimental groups ($n$ = 4 animals/group; HCC: 0.29% ± 0.3%; SI: 18.3% ± 3.9%, $p = 0.03$; Mann–Whitney test). Open circles are values from single animals and bars are mean ± SEM. *: $p < 0.05$; ***: $p < 0.001$; ****: $p < 0.0001$.

The online version of this article includes the following figure supplement(s) for figure 2:

**Source data 1.** Raw data of c-Fos quantification in the MSDB.

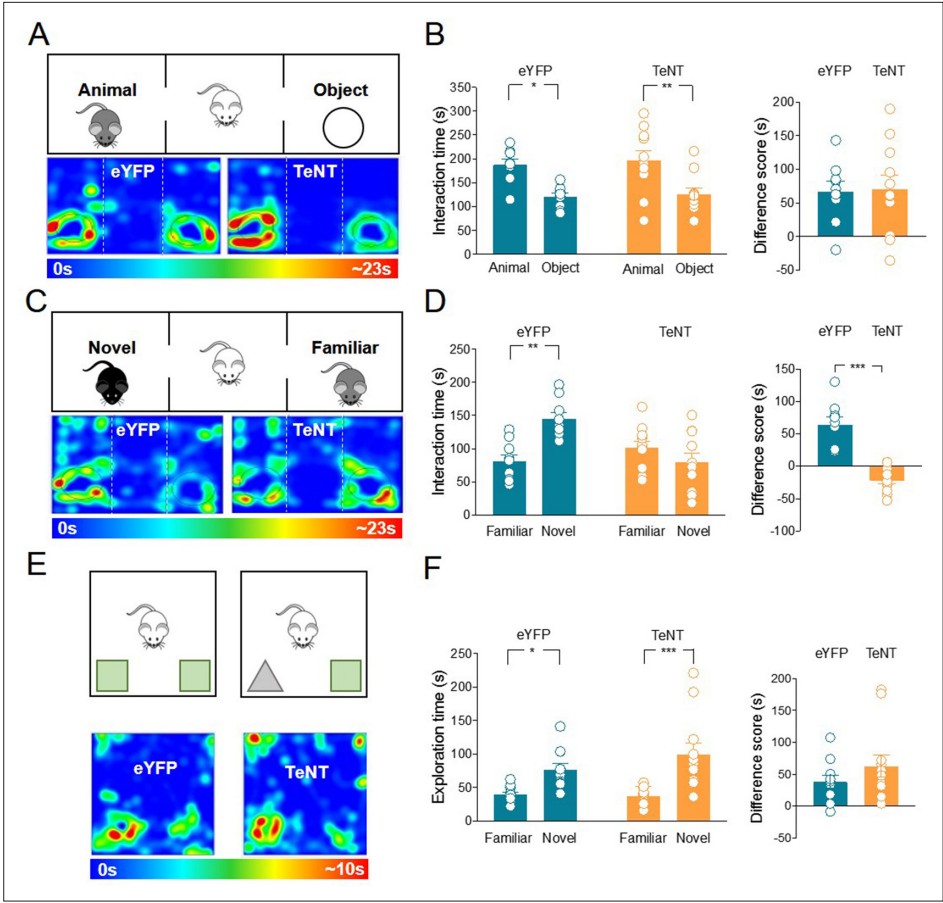

**Figure 3.** Inhibition of acetylcholine (ACh) release from medial septum/diagonal band of Broca (MSDB) ChAT[+] neurons impairs social novelty but not object recognition. (**A**) Top: schematic illustration of the sociability task in the three-chamber test. Bottom: representative heat map showing the time spent by an eYFP (left) or a TeNT (right) mouse in exploring the animal (left) and the object (right). (**B**) Left: aligned dot plot showing interaction time spent to explore the animal and the object during sociability task in eYFP (control, green, *n* = 9) and TeNT mice (orange, *n* = 11) (eYFP: 187 ± 12 vs 121 ± 7.7 s, p = 0.018; TeNT: 197 ± 20 vs 127 ± 12 s, p = 0.005; one-way ANOVA). Right: aligned dot plot showing the sociability score in eYFP (control, green, *n* = 9) and TeNT mice (orange, *n* = 11) (eYFP: 66.8 ± 15 s; TeNT: 70.5 ± 21 s, p = 0.94; Mann–Whitney test). (**C**) Top: schematic illustration of the social novelty task in the three-chamber test. Bottom: representative heat map showing the time spent by an eYFP (left) or a TeNT (right) mouse in exploring the novel animal (left) and the familiar one (right). (**D**) Left: aligned dot plot showing interaction time spent to explore the novel and the familiar animal in the social novelty task in eYFP (control, green, *n* = 9) and TeNT (orange, *n* = 11) mice (eYFP: 145 ± 10 vs 81.2 ± 10 s, p = 0.002; TeNT: 79.8 ± 13 vs 101 ± 10 s, p = 0.47; one-way ANOVA). Right: aligned dot plot showing the social novelty score in eYFP (control, green, *n* = 9) and TeNT (orange, *n* = 11) mice (eYFP: 64.1 ± 12 s; TeNT: −21.7 ± 5.8 s, p < 0.0001; Mann–Whitney test). (**E**) Top: schematic illustration of the novel object recognition (NOR) test. Bottom: representative heat map showing the time spent by an eYFP (left) or a TeNT (right) mouse in exploring the novel (left) and the familiar (right) object. (**F**) Left: aligned dot plot showing the exploration time spent to explore the familiar and the novel object during NOR task in eYFP (control, green, *n* = 11) and TeNT (orange, *n* = 12) mice (eYFP: 76.6 ± 8.8 vs 39 ± 3.6 s, p = 0.03; TeNT: 99.9 ± 16 vs 37.6 ± 9.8 s, p = 0.0002; one-way ANOVA). Right: aligned dot plot showing the exploration score in eYFP (control, green, *n* = 11) and TeNT (orange, *n* = 12) mice (eYFP: 37.6 ± 9.8 s; TeNT: 62.7 ± 17 s, p = 0.38; Mann–Whitney test). Open circles are values from single animals and bars are mean ± SEM. *: p < 0.05; **: p < 0.01; ***: p < 0.001.

The online version of this article includes the following figure supplement(s) for figure 3:

**Source data 1.** Interaction times and scores for three-chamber and novel object recognition tests.

**Figure supplement 1.** Intrinsic properties of TeNT-expressing ChAT[+] neurons in the medial septum/diagonal band of Broca (MSDB).

**Figure supplement 2.** TeNT expression in ChAT[+] neurons in the medial septum/diagonal band of Broca (MSDB) does not affect locomotor activity.

groups when analyzed during the execution of the social novelty task in the three-chamber apparatus (*Figure 3—figure supplement 2E-F*).

TeNT is persistently expressed for months after stereotaxic injection (*Kaspar et al., 2002*), thus inducing a long lasting block of ACh release from MSDB cholinergic neurons that could account for the observed effect on social novelty task. To rule out this hypothesis and to gain better temporal control of cholinergic inhibition, we sought to corroborate the results of the three-chamber test using the DREADD tool. Thus, an AAV carrying the floxed sequence of the human muscarinic receptor subtype four fused to the mCherry florescent protein (hM4) was injected in the MSDB. hM4 was engineered to be selectively activated by *N*-clozapine (CNO), a pharmacologically inert metabolite of the antipsychotic drug clozapine, inducing a hyperpolarization of the neuronal membrane via activation of a $G_i$-mediated potassium conductance (for review see *Sternson and Roth, 2014*). This effect relies on the time of CNO availability in the brain that lasts up to 200 min, depending on the concentration used and on the receptor desensitization rate (for review see *Roth, 2016*). hM4 was efficiently expressed by ChAT⁺ neurons in the MSDB (*Figure 4—figure supplement 1A*). Patch clamp recordings from hM4-expressing neurons in acute slices allowed to evaluate the effect of bath application of CNO (10 μM) on membrane potential and spontaneous firing frequency of ChAT⁺ neurons. CNO application hyperpolarized the membrane potential and reduced the firing frequency (*Figure 4—figure supplement 1B-C*). It was shown that in the absence of hM4, this CNO concentration does not alter the firing and membrane properties of hippocampal neurons (*Zhu et al., 2014*).

When tested for sociability in the three-chamber paradigm, hM4 mice exhibited a preference for the animal over the object as was the case for the control group (*Figure 4A-B*). As control group, we used eYFP mice treated with CNO to exclude possible side effects of this drug on social behavior that might result from its conversion into the psychoactive drug clozapine (*MacLaren et al., 2016*; *Gomez et al., 2017*). Before social novelty evaluation (30 min), CNO (3 mg/kg, dissolved in saline, for review see *Sternson and Roth, 2014*) was injected intraperitoneally (i.p.) in both eYFP and hM4 mice. This CNO concentration is effective in activating hM4 in mice without inducing unspecific behavioral effects due to clozapine conversion (*Jendryka et al., 2019*). Following CNO treatment, hM4 mice did not show the typical preference for the novel mouse compared to the familiar one as observed in the control group (*Figure 4C-D*). These data corroborate the results obtained with TeNT, overall indicating that ChAT⁺ neurons in the MSDB participate to, and are required for, social novelty discrimination.

We next sought to investigate whether social behavior-dependent activation of hippocampal CA2 neurons is affected by MSDB cholinergic inhibition. To this aim, animals expressing either control eYFP or hM4 were subjected to the three-chamber test and treated with CNO 30 min before the social novelty task. One hour later, animals were sacrificed and brain slices encompassing the CA2 region were immunolabeled for c-Fos. As shown in *Figure 4E-F*, social behavior induced activation of CA2 hippocampal neurons in eYFP control animals but not in hM4 mice. Interestingly, the majority of c-Fos⁺ neurons were PCP4⁻ (*Figure 4—figure supplement 2A-B*) suggesting a restricted activation of a subpopulation of PCP4⁻ principal neurons or local GABAergic interneurons. Immunolabeling with parvalbumin (PV), the marker of a subpopulation of GABAergic interneurons, revealed that the vast majority of c-Fos⁺ neurons was PV⁻ (*Figure 4—figure supplement 2A-C*).

To exclude the possibility that c-Fos activation was induced merely by the animals exploring the environment, we analyzed c-Fos expression in the CA2 region of animals exploring EA. We did not observe a significant difference between HCC and EA conditions (*Figure 4—figure supplement 3A-B*).

We next evaluated whether CNO-mediated inhibition of c-Fos expression in the CA2 of hM4 group that underwent SI extended to the neighboring CA3 and CA1 regions. No significant changes in c-Fos activation were observed in either CA3 or CA1 in mice of the three experimental groups (*Figure 4—figure supplement 4A-D*). Altogether, these data reveal that MSDB cholinergic neuron activation is elicited by social recognition tasks and is required for the downstream activation of the CA2 hippocampal region.

To assess whether cholinergic signaling is specifically relevant to social novelty discrimination, as compared to other forms of hippocampal-dependent cognitive tasks such as spatial novelty, we performed the object location test (OLT). During OLT, the test animal was first exposed to two identical objects. After 1 h, the animal was tested in the same arena with one of the two objects relocated to a new position (*Figure 4—figure supplement 5A*). hM4 and control eYFP animals received CNO

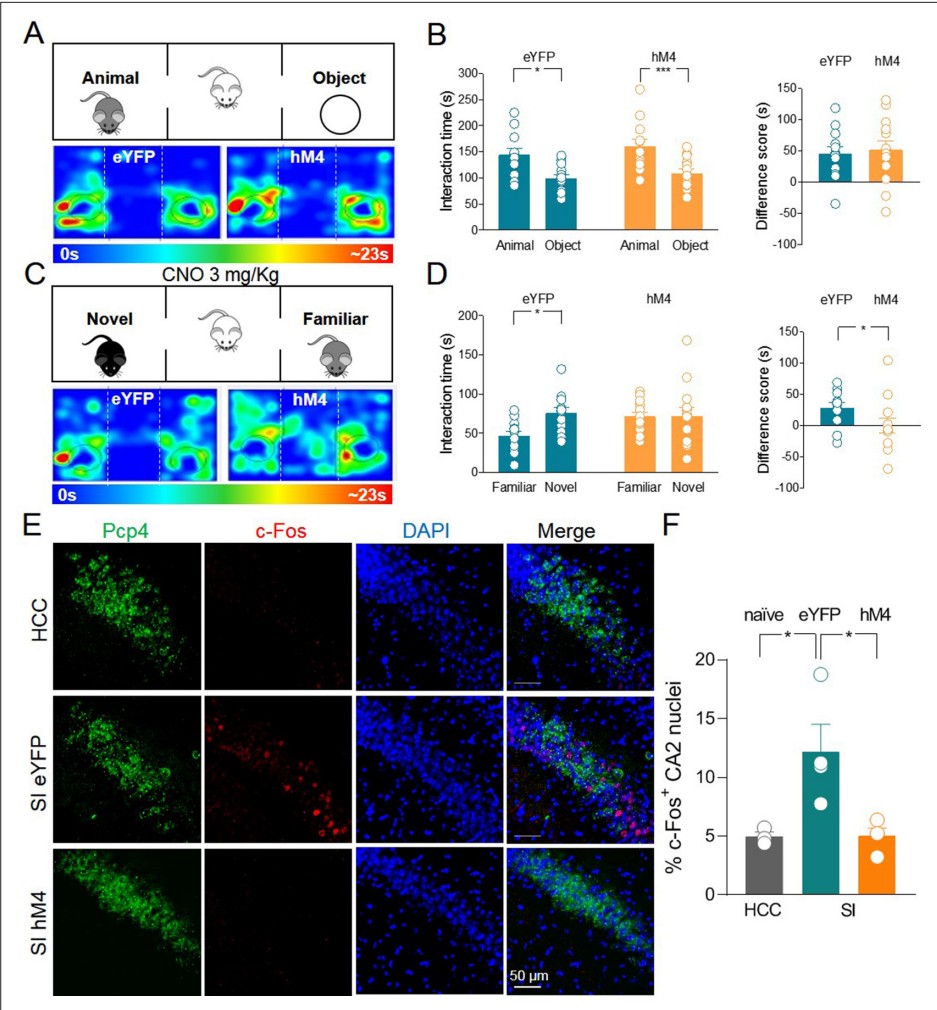

**Figure 4.** Silencing of hM4-expressing ChAT[+] neurons via systemic delivery of clozapine N-oxide (CNO) inhibits social novelty and c-Fos expression in the CA2 region. (**A**) Top: schematic illustration of the sociability task in the three-chamber test. Bottom: representative heat map showing the time spent by an eYFP (left) or a hM4 (right) mouse in exploring the animal (left) and the object (right). (**B**) Left: aligned dot plot showing interaction time spent to explore the animal and the object during sociability task in eYFP (control, green, $n$ = 12) and hM4 mice (orange, $n$ = 13) (eYFP: 144 ± 12 vs 99.1 ± 7.8 s, p = 0.02; hM4: 160 ± 13 vs 109 ± 8.2 s, p = 0.006; one-way ANOVA). Right: aligned dot plot showing the sociability score in eYFP (control, green, $n$ = 12) and hM4 mice (orange, $n$ = 13) (eYFP: 45 ± 12 s; hM4: 51 ± 15 s, p = 0.69; Mann–Whitney test). (**C**) Top: schematic illustration of the social novelty task in the three-chamber test performed 30 min after i.p. injection of CNO (3 mg/kg). Bottom: representative heat map showing the time spent by an eYFP (left) or a hM4 (right) mouse in exploring the novel (left) and the familiar (right) mouse. (**D**) Left: aligned dot plot showing interaction time spent to explore the novel and the familiar animal in the social novelty task in eYFP (control, green, $n$ = 12) and hM4 (orange, $n$ = 13) mice (eYFP: 75.5 ± 7.3 vs 46.8 ± 5.8 s, p = 0.01; hM4: 71.5 ± 11 vs 71.4 ± 5.4 s, p = 0.99; one-way ANOVA). Right: aligned dot plot showing the social novelty score in eYFP (control, green, $n$ = 12) and hM4 (orange, $n$ = 13) mice (eYFP: 28.7 ± 8.2 s; hM4: 0.15 ± 12 s, p = 0.03; Mann–Whitney test). (**E**) Confocal images showing (from left to right): Pcp4 (CA2 marker, in green), c-Fos (in red), nuclear-staining DAPI (blue), and merge images of CA2 hippocampal neurons. One hour after social novelty test, mice-expressing eYFP or hM4 in MSDB ChAT[+] neurons were sacrificed. The behavioral-dependent (social interaction, SI) activation of CA2 (detected by c-Fos immunostaining) was observed in eYFP mice but not in hM4 mice. Data were compared with those obtained from home-caged controls (HCCs). (**F**) Aligned dot plot showing the average percentage of c-Fos[+] nuclei detected in the CA2 region in the experimental groups ($n$ = 3–4 animals/group; HCC: 4.9% ± 0.4%; SI: 12.2% ± 2.3%; hM4: 5.02% ± 0.7%; p = 0.015, one-way ANOVA); open circles are values from single animals and bars are mean ± SEM. *:p < 0.05; ***: p < 0.001.

The online version of this article includes the following figure supplement(s) for figure 4:

**Source data 1.** Interaction times and scores for three-chamber and c-Fos quantification in CA2.

*Figure 4 continued on next page*

*Figure 4 continued*

**Figure supplement 1.** Clozapine N-oxide (CNO) application inhibits the spontaneous firing of hM4-expressing ChAT[+] neurons in the medial septum/diagonal band of Broca (MSDB).

**Figure supplement 2.** c-Fos activation of PCP4[−] neurons after social interaction.

**Figure supplement 3.** c-Fos activation in CA2 after empty arena exploration.

**Figure supplement 4.** The inhibition of medial septum/diagonal band of Broca (MSDB) ChAT[+] neurons does not affect c-Fos expression in the CA3 and CA1 regions.

**Figure supplement 5.** Silencing of hM4-expressing ChAT[+] neurons via systemic delivery of clozapine N-oxide (CNO) affects spatial memory.

i.p. injection 30 min before the test phase. hM4 mice showed a reduced interaction time with the object in the novel location as compared to the eYFP group (***Figure 4—figure supplement 5B***). These results suggest that the inhibition of MSDB cholinergic neurons also affects spatial novelty.

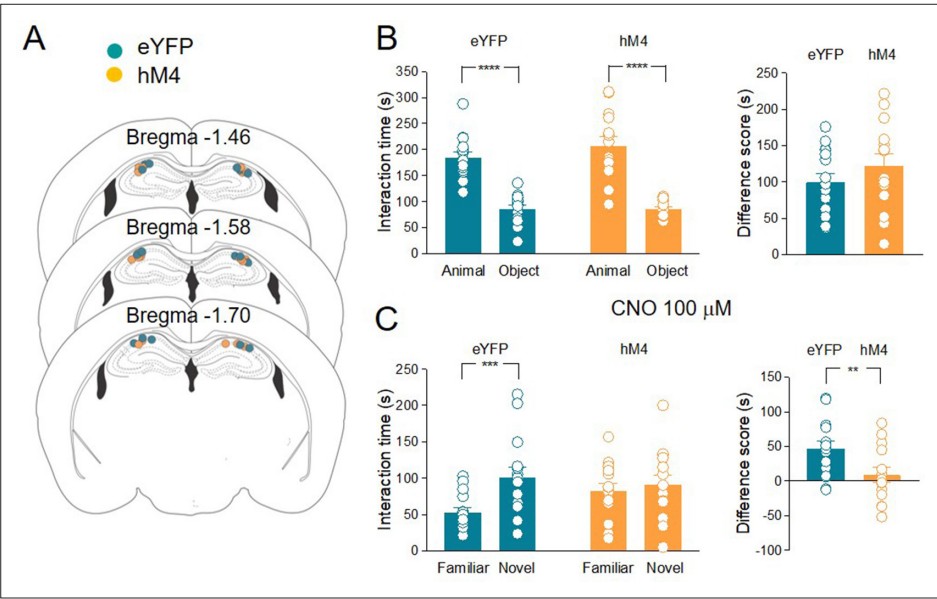

**Figure 5.** The inhibition of acetylcholine (ACh) release in the hippocampus impairs social novelty. (**A**) Schematic representations of cannula placements in the dorsal hippocampus of eYFP (control, green, *n* = 15) and hM4 (orange, *n* = 14) mice. (**B**) Left: aligned dot plots showing interaction time spent to explore the animal and the object during sociability task in eYFP (control, green, *n* = 15) and hM4 (orange, *n* = 14) mice (eYFP: 185 ± 10 vs 85.3 ± 7.3 s, p < 0.0001; hM4: 207 ± 17 vs 85 ± 3.7 s, p < 0.0001; one-way ANOVA). Right: aligned dot plot showing the sociability score in eYFP (control, green, *n* = 15) and hM4 (orange, *n* = 14) mice (eYFP: 99.4 ± 12 s; hM4: 122 ± 17 s, p = 0.27; Mann–Whitney test). (**C**) Left: aligned dot plots showing interaction time spent to explore the novel and the familiar animal in the social novelty task in eYFP (control, green, *n* = 15) and hM4 (orange, *n* = 14) mice following CNO (100 µM) delivery to the CA2 area of the hippocampus (30 min before social novelty task) (eYFP: 53 ± 6.6 vs 100 ± 14 s, p = 0.0009; hM4: 82.2 ± 10 vs 90.9 ± 13 s, p = 0.9; one-way ANOVA). Right: aligned dot plot showing the social novelty score in eYFP (control, green, *n* = 15) and hM4 (orange, *n* = 14) mice (eYFP: 47.4 ± 8.7 s; hM4: 8.7. ± 11 s, p = 0.3; Mann–Whitney test). Open circles are values from single animals and bars are mean ± SEM. **: p < 0.01; ***p < 0.001; ****: p < 0.0001.

The online version of this article includes the following figure supplement(s) for figure 5:

**Source data 1.** Interaction times and scores for three-chamber test.

**Figure supplement 1.** CA2 principal neuron properties.

**Figure supplement 2.** Clozapine N-oxide (CNO) affects both inhibitory and excitatory neurotransmission in CA2 from ChAT-Cre mice-expressing hM4 in medial septum/diagonal band of Broca (MSDB) neurons but not from naive mice.

## Local release of ACh in the CA2 hippocampal region is necessary for social novelty discrimination

It is known that cholinergic projections from MSDB nuclei target other brain areas such as the prefrontal cortex, the olfactory bulb, and the entorhinal cortex (*Li et al., 2018*; *Desikan et al., 2018*), which may also be involved in social novelty control. To test whether ACh release in CA2 is sufficient to control social novelty discrimination, CNO (100 µM) was locally applied through a cannula (*Figure 5A*) stereotactically implanted into the CA2 region. According to previous studies, the concentration of CNO for in vivo local application was tenfold higher than that used in slice recordings, which was effective in inducing neuronal silencing (*Meira et al., 2018*). Both eYFP (*n* = 15) and hM4 (*n* = 14) animal groups showed a preference for the animal rather than the object (*Figure 5B*). CNO was administered in the CA2 region 30 min before the social novelty test. In contrast to the control group (eYFP) hM4 animals did not show any significant difference in interaction time spent with the novel and the familiar mice, indicating that local release of ACh in the CA2 is necessary for social novelty discrimination (*Figure 5C*).

To unveil the mechanism by which ACh released in the hippocampus controls CA2 circuits, we performed whole-cell patch clamp experiments from CA2 pyramidal cells in hippocampal slices obtained from naive and ChAT-Cre mice-expressing hM4 in the MSDB. Firstly, we characterized the firing and membrane properties of the CA2 principal neurons. As shown in *Figure 5—figure supplement 1*, values similar to those previously published (*Chevaleyre and Siegelbaum, 2010*) were detected. Hence, pharmacologically isolated spontaneous inhibitory and excitatory postsynaptic currents (sIPSCs and sEPSCs) were recorded before and after CNO (10 µM) application in the presence of CNQX (10 µM), gabazine (10 µM), and physostigmine (3 µM) to block AMPA, GABA-A receptors, and acetylcholinesterase, respectively.

In slices obtained from naive mice not expressing hM4, bath application of CNO (10 µM) had no effect on either frequency or amplitude of sIPSCs or sEPSCs (*Figure 5—figure supplement 2A-D*). In contrast, in slices obtained from hM4 mice, CNO decreased the frequency, but not the amplitude of sIPSCs (*Figure 5—figure supplement 2E-F*). Furthermore, CNO decreased both the frequency and the amplitude of sEPSCs (*Figure 5—figure supplement 2G-H*). Possible off-target effects of CNO could be excluded since CNO differently affected the frequency and amplitude of synaptic currents recorded from naive and hM4 animals (*Figure 5—figure supplement 2I-J*). These data clearly demonstrate that ACh controls synaptic transmission in CA2.

## ACh controls social novelty discrimination via nAChR activation in CA2 neurons

Evidence has emerged that in the hippocampus and cortex ACh and GABA can be released from ChAT[+] fibers (*Takács et al., 2018*; *Saunders et al., 2015*; *Desikan et al., 2018*). To test whether ACh is sufficient to control social novelty discrimination, we selectively blocked ACh receptors locally. Specifically, nAChRs antagonist dihydro-β-erythroidine (DHβE) 50 mM or saline was administered in the CA2 area of the hippocampus in ChAT-Cre mice 30 min before the social novelty task (*Figure 6A-C*). The two groups of mice did not show differences in the sociability task (*Figure 6B*). However, when compared to saline-treated animals, DHβE-treated mice were impaired in the social novelty task (*Figure 6C*), indicating that nicotinic receptors are crucial for social novelty discrimination. In another set of experiments, mAChR antagonist atropine (1 mM) or saline was administered using the same paradigm of delivery (*Figure 6D-F*). Both groups of mice did not show differences in the sociability (*Figure 6E*) and social novelty tasks (*Figure 6F*), indicating that muscarinic receptor activation is not required for social novelty discrimination.

Analysis of locomotor activity in the three-chamber apparatus did not reveal any changes between eYFP and hM4 mice that received CNO injection (both i.p. and locally; *Figure 6—figure supplement 1A-B*), or between pharmacologically treated mice (i.e. DHβE and atropine, *Figure 6—figure supplement 1C-D*), excluding possible off-target effects of the drugs that were used.

## nAChR-mediated increase of glutamatergic transmission in CA2 principal cells via disinhibition

To understand how nicotinic ACh receptor activation impacts synaptic transmission in the CA2, we recorded spontaneous miniature inhibitory and excitatory postsynaptic currents (mIPSCs and mEPSCs)

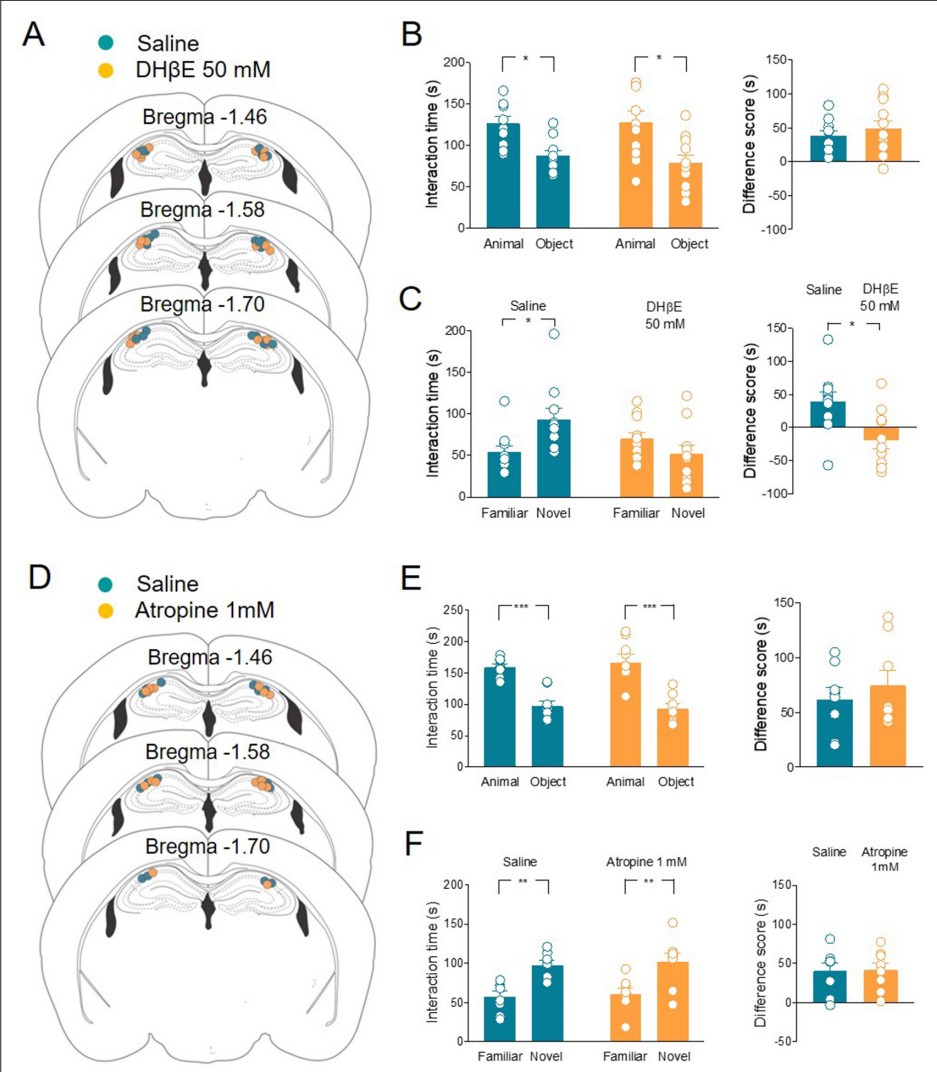

**Figure 6.** Local application of nAChRs, but not mAChRs antagonists in CA2 affects social novelty. (**A**) Schematic representations of cannula placements in the dorsal hippocampus of mice receiving a solution containing saline (control, green, $n = 10$) or dihydro-β-erythroidine (DHβE) (50 mM, orange, $n = 11$). (**B**) Left: aligned dot plots showing interaction time spent to explore the animal and the object during sociability task in the two groups of mice (green, $n = 10$ and orange, $n = 11$) (green: 126 ± 8 vs 87.3 ± 6.3 s, p = 0.03; orange: 128 ± 13 vs 79 ± 9 s, p = 0.005; one-way ANOVA). Right: aligned dot plot showing the sociability score in the two groups of mice (green, $n = 10$ and orange, $n = 11$) (green: 38.7 ± 7.3 s; orange: 48.8 ± 12 s, P = 0.6; Mann–Whitney test). (**C**) Left: aligned dot plot showing interaction time spent to explore the novel and the familiar animal in the social novelty task in saline- (control, green, $n = 10$) and DHβE-treated (orange, $n = 11$) mice (saline: 93.2 ± 14 vs 54.2 ± 7.7 s, p = 0.05; DHβE: 51.7 ± 10 vs 70.3 ± 7.5 s, p = 0.54; one-way ANOVA). Right: aligned dot plot showing the social novelty score saline- (control, green, $n = 10$) and DHβE-treated (orange, $n = 11$) mice (saline: 39 ± 15 s; DHβE: –18.6 ± 13 s, p = 0.016; Mann–Whitney test). (**D**) Schematic representations of cannula placements in mice that received saline solution (control, green, $n = 8$) or atropine (1 mM, orange, $n = 8$). (**E**) Left: aligned dot plot showing interaction time spent to explore the animal and the object during sociability task in the two groups of mice (green, $n = 8$ and orange, $n = 8$) (green: 159 ± 5.2 vs 97.3 ± 8.9 s, p = 0.0003; orange: 167 ± 14 vs 92.4 ± 8 s, p < 0.0001; one-way ANOVA). Right: aligned dot plot showing the sociability score in the two groups of mice (green, $n = 8$ and orange, $n = 8$) (green: 61.7 ± 11 s; orange: 74.4 ± 14 s, p > 0.99; Mann–Whitney test). (**F**) Left: aligned dot plot showing interaction time spent to explore the novel and the familiar animal in the social novelty task in saline- (control, green, $n = 8$) and atropine-treated (orange, $n = 8$) mice (saline: 97.9 ± 5.9 vs 57.4 ± 7.04 s, p = 0.009; atropine: 102 ± 11 vs 60.6 ± 7.6 s, p = 0.007; one-way ANOVA). Right: aligned dot plot showing the social novelty score in saline- (control, green, $n = 8$) and atropine-treated (orange, $n = 8$) mice (saline: 40.5 ± 10 s; atropine: 41.5 ± 9 s, p = 0.96; Mann–Whitney test). Open circles are values from single animals and bars are mean ± SEM. *: p < 0.05; **: p <

*Figure 6 continued on next page*

*Figure 6 continued*

0.01; \*\*\*: p < 0.001.

The online version of this article includes the following figure supplement(s) for figure 6:

**Source data 1.** Interaction times and scores for three-chamber test.

**Figure supplement 1.** Clozapine N-oxide (CNO) or cholinergic receptor antagonists did not affect locomotor activity within the social context (three chamber).

from CA2 principal neurons in acute hippocampal slices in the presence of tetrodotoxin (1 µM) to block sodium channels and action potentials propagation, CNQX (10 µM) or gabazine (10 µM) to block AMPA or GABA-A receptors, respectively (*Figure 7—figure supplement 1A-D*). Bath application of nicotine (1 µM) induced a significant decrease in the frequency of mIPSCs without affecting their amplitude (*Figure 7—figure supplement 1A-B*). No effect was observed in either frequency or amplitude of mEPSCs (*Figure 7—figure supplement 1C-D*), indicating that nAChR activation controls mainly GABAergic transmission in the CA2 region. These data are in agreement with previous studies showing a preferential expression of nAChRs by GABAergic interneurons (for review see *Griguoli and Cherubini, 2012*; *Pancotti and Topolnik, 2021*). To assess whether nicotine could indirectly control the activity of excitatory cells by acting on GABAergic interneurons, we recorded spontaneous action potential-dependent excitatory postsynaptic currents (sEPSCs) at the reversal potential of Cl⁻ ($E_{Cl^-}$ = −65 mV) without blocking GABAergic-mediated synaptic transmission. Under this condition, nicotine significantly increased the frequency of sEPSCs without affecting their amplitude (*Figure 7A-B*). In 6 out of 13 cells a partial recovery of the control frequency was obtained during nicotine washout (frequency: Nic: 1.45 ± 0.3 Hz, washout: 0.9 ± 0.1 Hz; p = 0.03; Wilcoxon test), ruling out a possible desensitizing effect of nicotine. These results suggest that nAChRs activation augments the glutamatergic drive to CA2 principal cells via disinhibition.

To identify the nAChR subtypes responsible for nicotine-induced disinhibition, we repeated the experiments in the presence of methyllycaconitine (MLA, 10 nM) or DHβE (0,5 µM) which, at these concentrations selectively block α7- and non-α7-nAChR subtypes, respectively. In the presence of either MLA or DHβE nicotine did not change the frequency of sEPSCs (*Figure 7C-F*), indicating that both α7- and non-α7-nAChR subtypes are involved in nicotine-induced CA2 disinhibition. To elucidate whether the effect of nicotine was due to nicotinic modulation of neighboring hippocampal areas, we recorded sEPSCs from CA3 and CA1 principal neurons. In contrast to what observed in CA2, in both CA3 and CA1 regions, nicotine significantly reduced the frequency of sEPSCs recorded at $E_{Cl^-}$ without blocking GABAergic transmission (*Figure 7—figure supplement 2A-D*). We then performed additional experiments to manipulate endogenous release of ACh via activation of the excitatory DREADD hM3, which leads to membrane depolarization when activated by CNO (*Alexander et al., 2009*). Patch clamp recordings from hM3-expressing neurons (*Figure 8—figure supplement 1A*) in acute slices allowed to evaluate the effect of CNO (10 µM) bath application on membrane potential and spontaneous firing of ChAT⁺ neurons. CNO application depolarized the membrane and increased the frequency of spontaneous action potentials (*Figure 8—figure supplement 1B-C*). We then evaluated the effect of CNO application on sEPSCs recorded from CA2 principal neurons in the presence of physostigmine (3 µM) and atropine (1 µM) to block the acetylcholinesterase and muscarinic receptors, respectively. In these conditions CNO increased the frequency, but not the amplitude of sEPSCs confirming the results obtained with nicotine (*Figure 8A-B*). In addition, as observed with nicotine, CNO decreased the frequency and the amplitude of sEPSCs recorded from CA3 principal neurons (*Figure 8C-D*). In contrast with results obtained using nicotine, CNO did not affect either the frequency or the amplitude of sEPSCs recorded from CA1 principal neurons (*Figure 8E-F*). This could be explained by a smaller number of cholinergic terminals expressing hM4 in CA1, as compared to CA2 and CA3 regions.

## Optogenetic activation of ChAT⁺ neurons in the MSDB increases the firing of CA2 neurons in vivo

To study the effects of endogenously released ACh on CA2 principal cells output in vivo, juxtacellular recordings were performed from neurons in the CA2 region of anesthetized (tiletamine/zolazepam-xylazine) ChAT-Cre mice, expressing ChR2-eYFP in the MSDB (*Figure 9A*). Principal cells were

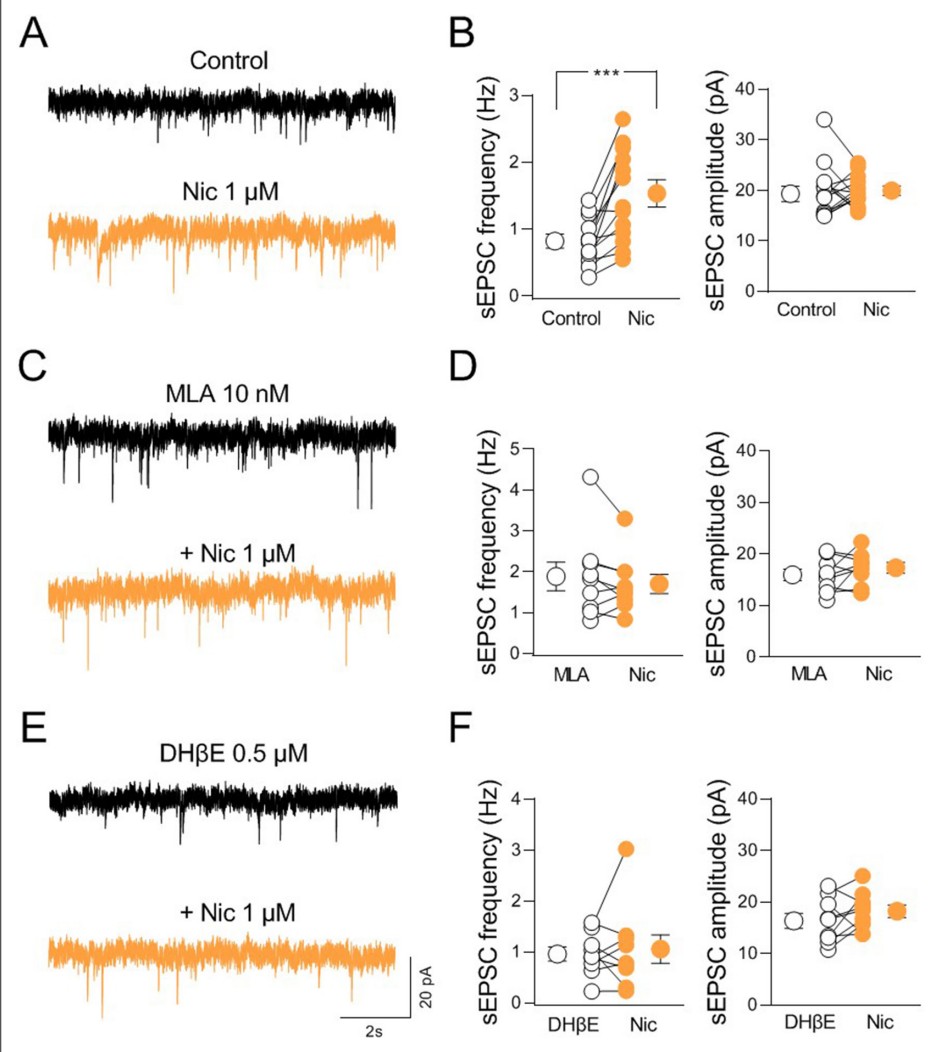

**Figure 7.** nAChR-mediated modulation of synaptic transmission in CA2. (**A**) Sample traces showing spontaneous excitatory postsynaptic currents (sEPSCs) of a CA2 pyramidal neuron recorded at $E_{Cl}^{-}$ in control (black) and in the presence of nicotine (1 μM; orange). (**B**) Aligned dot plots showing the mean frequency (left) and amplitude (right) of sEPSCs recorded from CA2 pyramidal neurons in control (black) and in the presence of nicotine (1 μM; orange) ($n$ = 13 cells; frequency: control: 0.81 ± 0.1 Hz, Nic: 1.5 ± 0.2 Hz; p = 0.0005; amplitude: control: 19.9 ± 1.4 pA, Nic: 19.7 ± 0.8 pA; p = 0.76; Wilcoxon test). (**C**) Sample traces showing sEPSCs of a CA2 pyramidal neuron recorded at $E_{Cl}^{-}$ in the presence of α7 nAChR antagonist methyllycaconitine (MLA, (10 nM; black)) and in the presence of MLA plus nicotine (1 μM; orange). (**D**) Aligned dot plots showing the mean frequency (left) and amplitude (right) of sEPSCs recorded from CA2 pyramidal neurons in the presence of MLA (10 nM; black) and in the presence of MLA plus nicotine (1 μM; orange) ($n$ = 9 cells; frequency: MLA: 1.9 ± 0.3 Hz, MLA+ Nic: 1.7 ± 0.2 Hz; p = 0.34; amplitude: MLA: 15.9 ± 1.1 pA, MLA+ Nic: 17.3 ± 1.0 pA; p = 0.36; Wilcoxon test). (**E**) Sample traces showing sEPSCs of a CA2 pyramidal neuron recorded at $E_{Cl}^{-}$ in the presence of non-α7 nAChR antagonist dihydro-β-erythroidine (DHβE, 0.5 μM; black) and in the presence of DHβE plus nicotine (1 μM; orange). (**F**) Aligned dot plots showing the mean frequency (left) and amplitude (right) of sEPSCs recorded from CA2 pyramidal neurons in the presence of DHβE (0.5 μM; black) or DHβE plus nicotine (1 μM; orange) (n = 9 cells; frequency: DHβE: 0.9 ± 0.1 Hz, DHβE + Nic: 1.05 ± 0.3 Hz; p > 0.99; amplitude: DHβE: 16.3 ± 1.5 pA, DHβE + Nic: 18.2 ± 1.2 pA; p = 0.25; Wilcoxon test). Open or closed circles represent values from single cells. Laterally located circles represent mean ± SEM. ***: p < 0.001.

The online version of this article includes the following figure supplement(s) for figure 7:

**Source data 1.** Frequency and amplitude of spontaneous excitatory postsynaptic currents .

**Figure supplement 1.** nAChR-mediated modulation of miniature synaptic events in CA2.

**Figure supplement 2.** nAChR-mediated modulation of synaptic transmission in CA3 and CA1 regions.

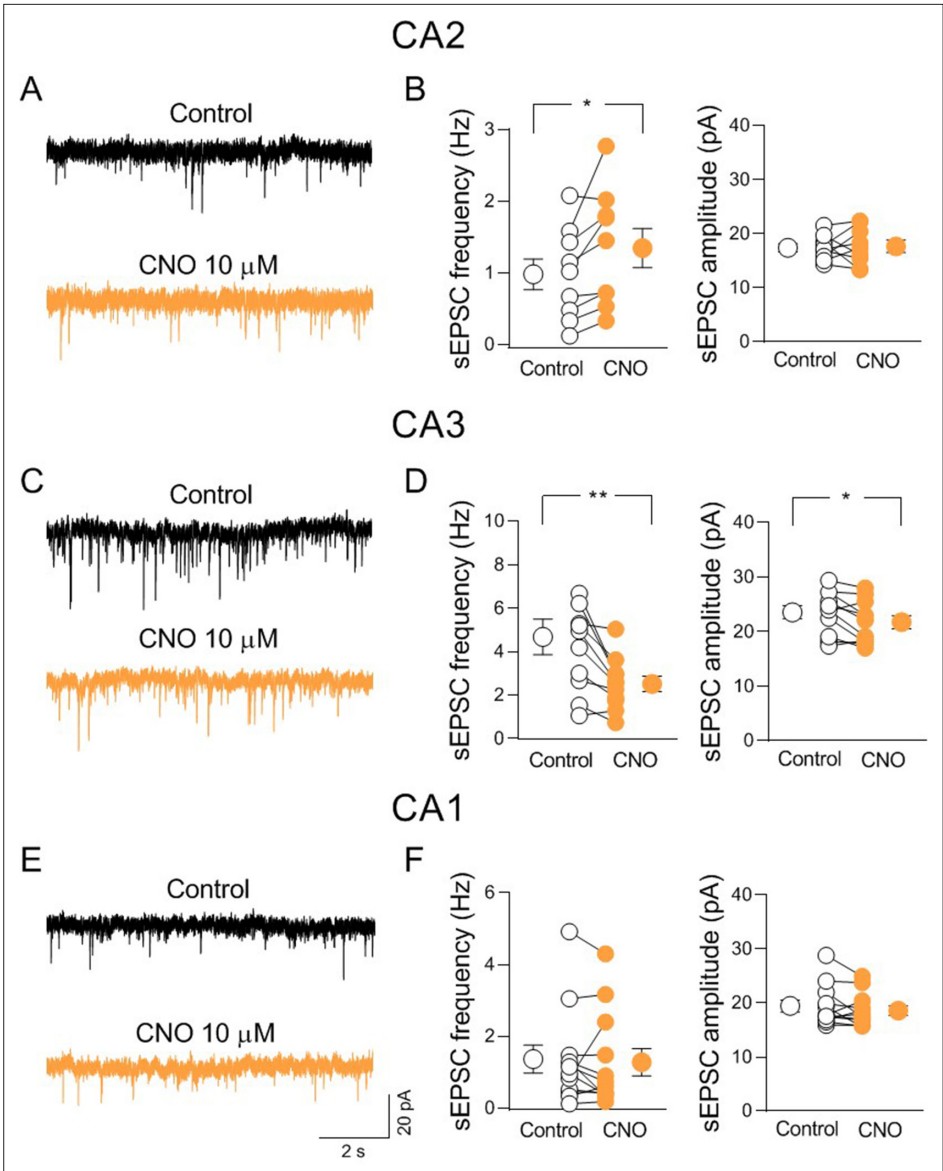

**Figure 8.** Activation of hM3-expressing cholinergic axon fibers modulates synaptic transmission in CA2 and CA3 but not CA1. (**A**) Sample traces showing spontaneous excitatory postsynaptic currents (sEPSCs) of a CA2 pyramidal neuron recorded at $E_{Cl}^-$ in control (black) and in the presence of clozapine N-oxide (CNO, 10 μM; orange). (**B**) Aligned dot plots showing the mean frequency (left) and amplitude (right) of sEPSCs recorded from CA2 pyramidal neurons in control (black) and in the presence of CNO (10 μM; orange) (n = 8 cells; frequency: control: 0.98 ± 0.2 Hz, CNO: 1.3 ± 0.3 Hz; p = 0.02; amplitude: control: 17.7 ± 0.8 pA, CNO: 17.7 ± 1.3 pA; p = 0.94; Wilcoxon test). (**C**) Sample traces showing sEPSCs of a CA3 pyramidal neuron recorded at $E_{Cl}^-$ in control (black) and in the presence of CNO (10 μM; orange). (**D**) Aligned dot plots showing the mean frequency (left) and amplitude (right) of sEPSCs recorded from CA3 pyramidal neurons in control (black) and in the presence of CNO (10 μM; orange) (n = 11 cells; frequency: control: 4.7 ± 0.8 Hz, CNO: 2.5 ± 0.3 Hz; p = 0.002; amplitude: control: 23.6 ± 1.2 pA, CNO: 21.7 ± 1.2 pA; p = 0.03; Wilcoxon test). (**E**) Sample traces showing sEPSCs of a CA1 pyramidal neuron recorded at $E_{Cl}^-$ in control (black) and in the presence of CNO (10 μM; orange). (**F**) Aligned dot plots showing the mean frequency (left) and amplitude (right) of sEPSCs recorded from CA1 pyramidal neurons in control (black) and in the presence of CNO (10 μM; orange) (n = 12 cells; frequency: control: 1.36 ± 0.4 Hz, CNO: 1.3 ± 0.4 Hz; p = 0.35; amplitude: control: 19.4 ± 1.1 pA, CNO: 18.6 ± 0.9 pA; p = 0.2; Wilcoxon test). Open or closed circles represent values from single cells. Laterally located circles represent mean ± SEM. *: p < 0.05; **: p < 0.01.

The online version of this article includes the following figure supplement(s) for figure 8:

**Source data 1.** Frequency and amplitude of spontaneous excitatory postsynaptic currents.

*Figure 8 continued on next page*

identified based on their bursting behavior (*Csicsvari et al., 1999*; *Ding et al., 2020*; *Figure 9B-C*). Brief light pulses delivered at low frequency (5 ms duration, at 1 Hz for 30 s) to cholinergic neurons in the MSDB by an optical fiber, increased the firing rate of CA2 neurons (*Figure 9B-D*) without altering the intraburst frequency (*Figure 9D*). This suggests that ACh release during light-induced ChR2 activation strongly affects the output of CA2 principal neurons. No effect was observed when light was delivered to not injected mice (*Figure 9—figure supplement 1A-D*). These data were independent from the anesthetic used, as a similar effect was observed when the animals were anesthetized with ketamine–xylazine (*Figure 9—figure supplement 1E-H*).

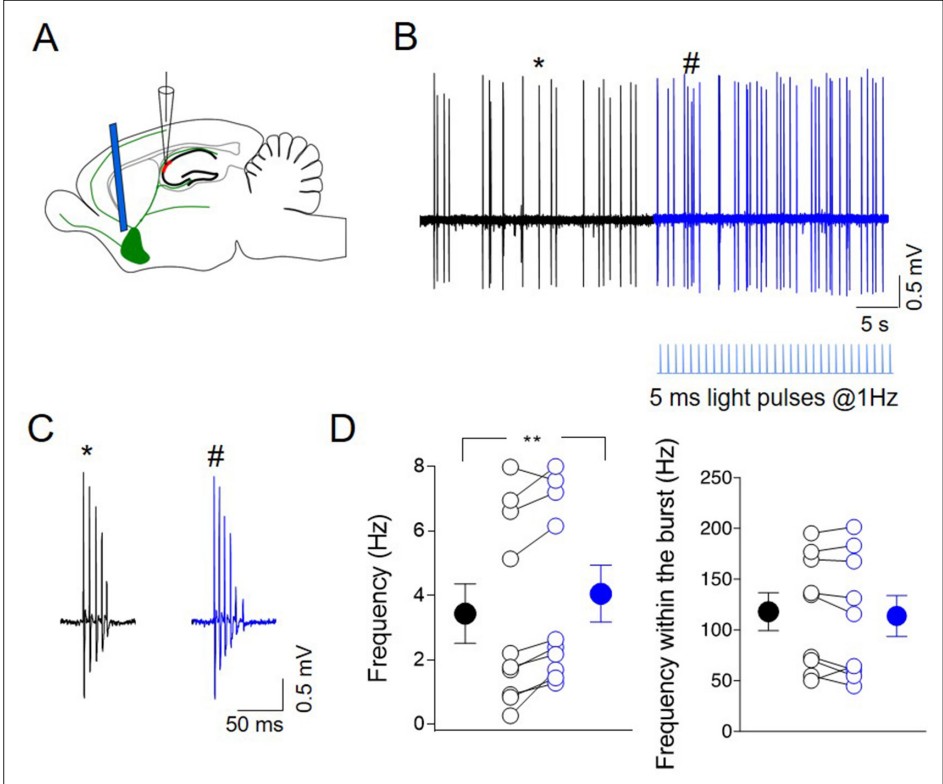

**Figure 9.** Photoactivation of ChAT[+] neurons in medial septum/diagonal band of Broca (MSDB) controls CA2 output. (**A**) Schematic illustration showing the experimental settings of in vivo juxtacellular recordings combined with light stimulation of MSDB ChAT[+] neurons expressing channelrhodopsin (ChR2). (**B**) Representative trace showing spontaneous firing from a CA2 bursting neuron in control (black) and during ChR2 activation (blue) via light pulses (below the trace). (**C**) Individual bursts in (**B**) (asterisk and hashtag for control and light activation, respectively) shown on an expanded time scale. (**D**) Left: aligned dot plot showing the frequency of spikes in control (black) and during light activation of ChAT[+] neurons in the MSDB (blue) (*n* = 9 cells; control: 3.62 ± 1.0 Hz; light: 4.26 ± 0.97 Hz; p = 0.012, Wilcoxon test); right: aligned dot plot showing the frequency of spikes within the bursts in control and during light activation of ChAT[+] neurons in the MSDB (*n* = 9 cells; control: 118 ± 19 Hz; light: 114 ± 20 Hz; p = 0.36, Wilcoxon test). Open circles represent values from single cells. Closed circles represent mean ± SEM. **: p = 0.01.

The online version of this article includes the following figure supplement(s) for figure 9:

**Source data 1.** Spike analysis.

**Figure supplement 1.** Light delivery in MSDB combined with CA2 recordings in not injected or ChR2-expressing mice under ketamine–xylazine anesthesia.

**Figure supplement 2.** Photoactivation of ChAT[+] neurons in MSDB does not affect CA3 and CA1 output.

Additional experiments were performed to study the effect of MSDB light stimulation on both CA3 and CA1 regions. Photostimulation of ChAT[+] neurons in the MSDB induced no significant change in the firing frequency of both CA3 and CA1 bursting cells (*Figure 9—figure supplement 2A-H*).

In line with slices recordings, these in vivo data suggest that ACh released from MSDB enhances CA2 principal cells firing via disinhibition, by activating nAChRs localized on GABAergic interneurons.

## Discussion

Here, we show that cholinergic inputs from the MSDB support social memory mediated by the CA2 region of the hippocampus. Among social areas in the brain, the CA2 has recently emerged as a key structure for social cognition (for review see *Dudek et al., 2016*; *Piskorowski and Chevaleyre, 2018*). *Hitti and Siegelbaum, 2014* clearly demonstrated that genetically targeted inactivation of CA2 principal cells leads to a loss of social memory, namely the ability of an animal to recognize a conspecific. More recently, evidence was provided that consolidation of social memory strictly depends on reactivation of CA2 pyramidal cell ensembles during sharp-wave ripples (*Oliva et al., 2020*). However, despite the increasing information regarding the circuits that are involved, the underlying mechanisms are still largely unknown. Among possible candidates taking part in social cognition, cholinergic signaling may play a key role. This signaling pathway is known to be involved in several cognitive processes (for review see *Ballinger et al., 2016*; *Solari and Hangya, 2018*), and deficits in cholinergic transmission are associated with cognitive impairments in various forms of neuropsychiatric disorders (for review see *Dineley et al., 2015*; *Terry and Callahan, 2020*). Our results indicate that social memory requires the activation of cholinergic neurons in the MSDB. In particular, c-Fos immunolabeling revealed a selective activation of ChAT[+] neurons in response to social stimuli. Furthermore, the inhibition of cholinergic neurons in the MSDB with TeNT severely impaired social novelty discrimination, indicating that ACh is involved in this task. Notably, this effect was not related to the novelty per se since TeNT-expressing mice did not show an impairment in the NOR test. This result is in agreement with findings demonstrating that an increased level of ACh in the hippocampus during object exploration was not related to object familiarity or novelty (*Stanley et al., 2012*). Furthermore, ACh released in the perirhinal cortex seems to play a major role in the novel object discrimination task by acting via mAChRs (*Winters et al., 2006*; *Balderas et al., 2012*) rather than nAChRs (*Tinsley et al., 2011*).

Social memory impairment induced by TeNT was also observed in hM4-expressing mice treated with CNO, known to block the firing of targeted neurons via membrane hyperpolarization. These results strongly support the role of ACh in social novelty discrimination. However, ACh seems not to play an exclusive role in social novelty discrimination as hM4 mice, subjected to OLT, also showed an impairment in spatial novelty discrimination. This result is in agreement with a previous study, whereby ChAT[+] neurons in the MSDB were selectively eliminated using a genetic cell targeting technique (*Okada et al., 2015*). Deficits in OLT were also observed in mice lacking the M2 mAChR subtype suggesting the involvement of these receptors in this task (*Romberg et al., 2018*).

We focused on the role of cholinergic signaling in the CA2 region that is key to social memory formation and is highly innervated by cholinergic fibers originating from ChAT[+] cells in the MSDB. Analysis of c-Fos expression following social behavior unveiled an increased learning-dependent activation of neurons in the CA2 region, which was reduced upon hM4-dependent inhibition of MSDB cholinergic neurons. Previous studies (*Wintzer et al., 2014*; *Alexander et al., 2016*) demonstrated that exposure to a novel context elicits learning-dependent transcription of the immediate-early gene *Arc* in the CA2 region of rodents. In our experiments, animals exposed to the three-chamber arena in the absence of social stimuli did not show a significant increase in c-Fos expression in the CA2 region. However, we cannot state that the increase in c-Fos staining in CA2 is selective for social stimuli since a similar response to a novel object has not been assessed (*Alexander et al., 2016*).

The notion that social memory depends on MSDB cholinergic inputs to the CA2 region relies on evidence that local CNO-triggered inhibition of cholinergic fibers significantly impaired this task. Although the volume of drugs injected locally in CA2 was very small, the possibility of spillover to adjacent CA3 and CA1 areas cannot be excluded. Hence, this finding warrants further support that ideally would be based on another experimental approach. Furthermore, we provided evidence that social memory requires the activation of nicotinic, but not muscarinic AChRs. Previous work suggested the involvement of nAChRs in social interactions. In particular, the lack of β2 nAChR subtype was shown to affect social interaction during aggressive behavior (*Granon et al., 2003*), an effect likely

involving the medial prefrontal cortex (mPFC; *Avale et al., 2011*). The lack of β4 nAChR subunit that is highly expressed in the olfactory bulb and in the lateral habenula (*Salas et al., 2003*), leads to a decreased interaction between a resident and juvenile intruder mouse (*Salas et al., 2013*). Furthermore, a single nucleotide polymorphism in the α5 subunit of nAChRs, observed in patients affected by schizophrenia (*Schizophrenia Working Group of the Psychiatric Genomics Consortium, 2014*), leads to deficits in sociability when engineered in mice (*Koukouli et al., 2017*).

The effect of nAChRs activation on glutamatergic and GABAergic terminals impinging on CA2 pyramidal cells is not known. Our ex vivo patch clamp recordings of miniature synaptic currents from CA2 principal cells in hippocampal slices did not unveil any effects of nicotine on spontaneous glutamatergic events in comparison to GABAergic ones, suggesting that ACh receptors are preferentially expressed on GABAergic interneurons. nAChR-mediated modulation seems to be region specific since the activation of nAChRs increases GABA release in CA1 (*Rosato-Siri et al., 2006*; *Tang et al., 2011*) that may account for the observed reduction in the frequency of sEPSCs. Similarly, in CA3 principal cells, activation of nAChRs by low concentration of nicotine enhances GABA release either directly (*Hajós et al., 2005*) or indirectly, via NMDA receptors localized on GABAergic interneurons (*Mann and Mody, 2010*; *Wang et al., 2015*). Furthermore, previous studies showed that brain states associated with high cholinergic activity and theta oscillations resulted in CA3 principal neuron inhibition via activation of interneurons (*Malezieux et al., 2020*; *Dannenberg et al., 2015*). Thus, enhanced GABAergic tone may lead, as in CA1, to a reduced glutamate release from CA3 principal cells. Our results on nicotine-mediated control of glutamatergic transmission in CA2, CA3, and CA1 regions were confirmed by ACh released from hM3-expressing hippocampal fibers, supporting a circuit-specific regulation of CA2 by nAChRs.

The inhibitory effect of CNO on spontaneous EPSCs and IPSCs in mice-expressing hM4 in MSDB ChAT+ neurons may be related to its action on both nicotinic and muscarinic receptors, the latter known to be present on CA2 neurons (*Robert et al., 2020*).

In our experiments, the decrease of GABAergic transmission induced by nicotine in CA2 was associated with an indirect increase in frequency of spontaneous glutamatergic events, recorded at the equilibrium potential for chloride, suggesting a disinhibitory mechanism. Interestingly a disinhibitory effect of ACh was also observed in the prelimbic area of mPFC, where α5 nAChR-dependent activation of VIP+ interneurons inhibited downstream SOM+ cells, which in turn led to an enhancement of principal neuron firing in layer 2/3 (*Koukouli et al., 2017*). In accordance with results based on slice recordings, our in vivo experiments showed that optogenetic activation of MSDB ChAT+ neurons enhanced the firing frequency of CA2 pyramidal neurons identified by their bursting behavior (*Csicsvari et al., 1999*; *Ding et al., 2020*), indicating that ACh strongly controls the CA2 output. This seems to be selective for the CA2 region, since optogenetic activation of MSDB ChAT+ neuron did not modify the firing frequency of neighboring CA3 and CA1 putative principal neurons. The lack of an effect in CA3 and CA1 may be related to the concomitant activation of mAChRs, probably differently distributed among hippocampal areas. Altogether, our results are commensurate with a scenario in which ACh controls social memory through nAChR activation by decreasing GABAergic signaling and thus enhancing the excitatory drive to CA2 principal cells. Recordings from GABAergic interneurons will be critical to identify how different subtypes contribute to social memory via nAChR-mediated disinhibition.

## Materials and methods

### Animals

All experiments were performed in accordance with the Italian Animal Welfare legislation (D.L. 26/2014) that were implemented by the European Committee Council Directive (2010/63 EEC) and were approved by local veterinary authorities, the EBRI ethical committee and the Italian Ministry of Health (565/PR18). All efforts were made to minimize animal suffering and to reduce the number of animals used. At least four to five male mice were used for a given experiment. We used *B6;129S6-Chat tm2(cre)Lowl/J* (ChAT-Cre), purchased from Jackson Laboratory (Stock No: 006410). Experiments were performed on male off-spring derived from homozygous mating. Only male mice were used in this study to limit the effects of estrous cycle on cholinergic signaling (*Gibbs, 1996*). Mice were housed in 4–5 per cage at constant temperature (22°C) and humidity (30–50%) and were kept on a

regular circadian cycle (12 h:12 h light:dark cycle, lights on at 7:00 a.m.). Mice were provided with food and water ad libitum.

## Viruses

Adeno-associated virus (AAV) containing Tetanus toxin light chain [AAV-DJ CMV DIO eGFP-2A-TeNT] was purchased from Stanford University Gene Vector and Virus core (CA, USA) with genomic titer of $1.4 \times 10^{13}$ particles/ml. AAV serotype 2/9 containing channelrhodopsin-2 [ChR2; AAV-DIO-ChR2(H134R)-enhanced yellow fluorescent protein (eYFP) or -mCherry] or AAV-Ef1a-DIO eYFP with genomic titers of $1.49 \times 10^{13}$ and $3.95 \times 10^{13}$ particles/ml, respectively, AAV serotype 2/8 containing human muscarinic receptor 3 and 4 DREADD [hM3D(Gq) and hM4D(Gi); pAAV-hSyn-DIO-hM3D(Gq)-mCherry and pAAV-hSyn-DIO-hM4D(Gi)-mCherry] with genomic titers of $4.0 \times 10^{12}$ particles/ml and $1.4 \times 10^{13}$ particles/ml were purchased from Addgene (MA, USA).

## In vivo stereotactic injections

ChAT-Cre mice (postnatal (P) day P25–30; weight 16–20 g) were anesthetized with an intraperitoneal injection (i.p.) of mixture of tiletamine/zolazepam (zoletyl, 80 mg/kg body weight) and xylazine (rompun, 10 mg/kg body weight). Viral vectors were injected in the MSDB (anteroposterior [AP], 0.8 mm; mediolateral [ML], 0.5 mm; dorsoventral [DV], −4.5 mm; all coordinates were relative to Bregma) through a 26 G needle lowered at an angle of 6.5° (in the ML axis) relative to the vertical plane in order to avoid sagittal sinus (*Boyce et al., 2016*). The injection volume and flow rate (500 nl at 60 nl/min) were controlled with an injection pump (UMP3 UltraMicroPump, World Precision Instruments, USA). After a minimum of 4 weeks that allowed for protein expression, mice were used for experiments.

## Cannula implantation for local drug delivery

Guide cannulas were placed at specific coordinates for targeting the dorsal hippocampal CA2. To this aim, a small hole was drilled bilaterally following the coordinates: AP = −1.6 mm, ML = ±1.7 mm, DV = −1.3 mm (all coordinates were relative to Bregma). Bilateral guide cannulas (7 mm length, 0.5 mm outer diameter with 0.25 mm inner diameter; Unimed, Switzerland) were slowly lowered into the brain through the holes until the target DV coordinate was reached. Quickly, the cannulas were fixed with acrylic dental cement (Riccardo Ilic, Italy) to be stably held on the calvarium at the DV coordinate established. Mice were let to recover from surgery for 1 week.

## Drug and injection procedure

Clozapine N-oxide (CNO, 100 µM), DHβE (50 mM) or atropine (1 mM), all dissolved in saline, were injected bilaterally (150 nl/hemisphere, 75 nl/min injection rate) in the dorsal hippocampus (CA2) 30 min before the social novelty test. To this end, mice were gently restrained to insert one of the guide cannulas, the injection needle (length, 7.6 mm; diameter, 0.25 mm) connected with a plastic tube to a 2 µl Hamilton syringe (Hamilton 7002 N – G 0.5/70 mm/pts2; Hamilton Company, USA). The needle was left in place for an additional 1 min to allow diffusion of the preparation. Immediately after, the injection was repeated in the other hemisphere using the same procedure. During the injections, mice were awake and free to move in the holding cage. CNO injected in mice-expressing control virus (eYFP) was used as control, ACSF was used as vehicle in the experiments with DHβE and atropine.

## Behavioral experiments

### Three-chamber test

Sociability and social novelty skills were tested using the three-chamber test, adapted from *Moy et al., 2004* in a homemade rectangular, clear Plexiglas three-chambered box (each chamber was 20 × 40 × 21 cm in size). Dividing walls included rectangular openings (6 × 8.5 cm) allowing access to each chamber. The light intensity (6 lux) was distributed equally in the apparatus. Between trials, the chambers of the arena were cleaned with 70% ethanol (EtOH) to eliminate lingering smells. Mice were handled 5 min a day for 5 days before the test. On the day before the test, mice were habituated to the empty chamber for 30 min. On the test day, after a 10 min habituation phase in the empty apparatus, during the first two trials (sociability task) the test mouse was placed in the middle compartment and allowed to explore for 10 min between a wire cup (ø 10.5 cm × 10.5 cm h) with an unfamiliar

juvenile (P30) C57BL7/6J male mouse (stranger one) and an identical empty wire cup. The position of stranger one was alternated between the first and second trial, to prevent side preference. The interaction time was recorded by the video-tracking system (ANY-maze, StoeltingCo, IL, USA), and the score was calculated as the difference between the investigation time for the novel mouse and that for the empty cup. After a 1 h intersession interval a second unfamiliar C57BL7/6J male mouse (stranger two or novel) was placed into the previously empty wire cup, while stranger one (familiar) remained inside its cup. The subject mouse was given 10 min to explore all three chambers. The score for social novelty was calculated as the difference between the investigation time for the novel mouse and that for the familiar mouse.

## Open-field exploration test

Open field was used to test general locomotor activity, anxiety, and willingness to explore. The experimental apparatus consisted of a black rectangular open field (60 × 60 × 30 cm; Panlab, USA). The arena was cleaned with 70% EtOH between trials to eliminate lingering smells. During the test, each animal was placed in the center of the arena and allowed to freely move for 10 min while being recorded with an overhead camera. The mouse activity was then analyzed by an automated tracking system (ANY-maze, StoeltingCo, IL, USA) for the following parameters: total ambulatory distance and velocity. A series of 12 × 12 cm zones were identified and used to evaluate the time spent in the center (inner zone) or peripheral zones (outer zone). The outer zone consisted of 12 blocks close to walls while the inner zone consisted of 9 blocks in the center. Greater time spent in the outer zones of the maze was indicative of amplified anxiety-related behavior (*Seibenhener and Wooten, 2015*).

## NOR test

The NOR test was slightly adapted as described in *Leger et al., 2013*. The experimental apparatus consisted of a black rectangular open field (60 × 60 × 30 cm; Panlab, USA). Prior to training, mice were handled for 5 min a day for 5 days. On the day before the test, mice were placed in the empty chamber and allowed to explore for 30 min. During the training phase mice were placed in the experimental apparatus in the presence of two identical objects and were allowed to explore for 10 min. After 1 h, mice were placed again for 10 min in the apparatus, where one of the objects had been replaced by a novel one. The objects consisted of two plastic boxes with different shapes, both approximately of the same height. In both tasks, the arena was cleaned with 70% EtOH to eliminate lingering smells. The interaction time with the familiar object and the novel one was recorded by the video-tracking system (ANY-maze, Stoelting Co, IL, USA) and manually analyzed.

## Object location test

Animals were tested by using OLT adapted from *Denninger et al., 2018*. This test is based on the spontaneous tendency of rodents to recognize when an object has been relocated. The experimental apparatus and the objects were the same used in the NOR task. Visual cues were inserted into the arena to help mice in spatial orientation. Prior to training, mice were handled for 5 min a day for 5 days. The day before the test, mice were placed in the empty chamber, and allowed to explore for 30 min. During the training phase mice were placed in the experimental apparatus in the presence of two identical objects and allowed to explore for 10 min. After 1 h, the location of one object was changed and the animal was free to explore the two objects for 10 min. In both tasks, the arena was cleaned with 70% EtOH to eliminate lingering smells. The interaction time with the familiar location and the novel location was recorded by the video-tracking system (ANY-maze, Stoelting Co, IL, USA) and manually analyzed.

## Slice preparation

Transverse hippocampal slices (320 μm tick) were obtained from P60–P70-old animals, using a standard protocol (*Bischofberger et al., 2006*). Briefly, after being anesthetized with an intraperitoneal injection of a mixture of tiletamine/zolazepam (zoletyl, 80 mg/kg body weight) and xylazine (rompun, 10 mg/kg body weight), mice were decapitated. The brain was quickly removed from the skull, placed in artificial cerebrospinal fluid (ACSF) containing (in mM): sucrose 75, NaCl 87, KCl 2.5, $NaH_2PO_4$ 1.25, $MgCl_2$ 7, $CaCl_2$ 0.5, $NaHCO_3$ 25, and glucose 25. After recovery, an individual slice was transferred to a submerged recording chamber and continuously perfused at room temperature with oxygenated

ACSF at a rate of 3 ml/min. ASCF saturated with 95% O2 and 5% CO2 and contained in mM: NaCl 125, KCl 2.5, NaH$_2$PO$_4$ 1.25, MgCl$_2$ 1, CaCl$_2$ 2, NaHCO$_3$ 25, and glucose 10.

## Electrophysiological recordings in slices

Cells were visualized with a 60 × water immersed objective mounted on an upright microscope (Nikon, eclipse FN1) equipped with a CCD camera (Scientifica, UK). Whole-cell patch clamp recordings, in voltage and current clamp modes, were performed with a MultiClamp 700B amplifier (Axon Instruments, Sunnyvale, CA, USA). Patch electrodes were pulled from borosilicate glass capillaries (WPI, Florida, US); they had a resistance of 3–4 MΩ when filled with an intracellular solution containing (in mM): K gluconate 70, KCl 70, HEPES 10, EGTA 0.2, MgCl$_2$ 2, MgATP 4, MgGTP 0.3, Na-phosphocreatine 5; the pH was adjusted to 7.2 with KOH; the osmolarity was 295–300 mOsm. Membrane potential values were not corrected for liquid junction potentials. Miniature GABA$_A$-mediated inhibitory postsynaptic currents (mIPSCs) and AMPA-mediated excitatory postsynaptic currents (mEPSCs) were recorded in the CA2 region of the hippocampus from a holding potential of −70 mV in the presence of tetrodotoxin (TTX 1 µM) and 1 6-cyano-7-nitroquinoxaline-2,3-dione (CNQX, 10 µM) or gabazine (10 µM), respectively. In patch clamp experiments performed to record spontaneous AMPA-mediated excitatory postsynaptic currents (sEPSCs) the electrodes were filled with an intracellular solution containing: K gluconate 127, KCl 6, HEPES 10, EGTA 1, MgCl$_2$ 2, MgATP 4, MgGTP 0.3; the pH was adjusted to 7.2 with KOH; the osmolarity was 290–300 mOsm. sEPSCs were recorded at the equilibrium potential for chloride (E$_{Cl-}$) that was approximately −65 mV based on the Nernst equation. Membrane potential values were not corrected for liquid junction potentials. The stability of the patch was checked by repetitively monitoring the input and series resistance during the experiments. Series resistance (10–20 MΩ) was not compensated. Cells exhibiting 15% changes were excluded from the analysis. Drugs were applied in the bath and the ratio of flow rate to bath volume ensured complete exchange within 2–3 min.

## Data analysis

Data were transferred to a computer hard disk after digitization with an A/D converter (Digidata 1550, Molecular Devices, Sunnyvale, CA, USA). Data acquisition (digitized at 10 kHz and filtered at 3 kHz) was performed with pClamp 10.4 software (Molecular Devices, Sunnyvale, CA, USA). Input resistance and cells capacitance were measured online with the membrane test feature of the pClamp software. Spontaneous and miniature EPSCs and IPSCs were analyzed with pClamp 10.4 (Molecular Devices, Sunnyvale, CA, USA). This program uses a detection algorithm based on a sliding template. The template did not induce any bias in the sampling of events because it was moved along the data trace by one point at a time and was optimally scaled to fit the data at each position.

## Drugs

Drugs were applied in the bath by gravity by changing the superfusion solution to one differing only in its content of drug(s). The following drugs were used: CNQX, DL-APV, and picrotoxin purchased from Tocris (UK), nicotine (SML1236), atropine, and DHβE purchased from Sigma (USA), CNO purchased from Abcam (UK), physostigmine hemisulfate purchased from Santa Cruz (USA). Stock solutions were made in distilled water and then aliquoted and frozen at −20 °C. Picrotoxin and CNQX were dissolved in dimethyl sulfoxide (DMSO). The final concentration of DMSO in the bathing solution was 0.1%. At this concentration, DMSO alone did not modify the membrane potential, input resistance, or the firing properties of CA2 neurons.

## Electrophysiological recordings in vivo

Mice (P60–70) were anesthetized with i.p. injection of a mixture of tiletamine/zolazepam (zoletyl; 80 mg/kg) and xylazine (rompun, 10 mg/kg body weight ) to induce anesthesia before surgery and during recordings. A subset of animals was injected with i.p. injection of a mixture of ketamine (lobotor; 100 mg/kg body weight) and xylazine (rompun, 10 mg/kg body weight). Temperature was maintained between 36°C and 37°C using a feedback-controlled heating pad. Two craniotomies for optogenetic stimulation (MSDB) and for recording (CA2) sites were drilled at +0.8 mm AP and +0.5 L for MSDB and −1.6 mm AP and +1.7 L for CA2 all relative to Bregma. Extracellular mapping allowed to locate the depth of CA2 pyramidal cell layer. Extracellular recordings of field potentials for activity mapping

were obtained with glass electrodes (Hingelberg, Malsfeld, Germany) prepared with a vertical puller PP-830 (Narishige, Japan), and the tip was broken to obtain a resistance between 1 and 2 MΩ. Electrodes were filled with a standard Ringer's solution containing the following (in mM): NaCl 135, KCl 5.4, HEPES 5, CaCl$_2$ 1.8, and MgCl$_2$ 1.

Juxtacellular recordings of spontaneous neuronal firing were obtained using glass electrodes (7–10 MΩ) filled with potassium-based solution. Activation of ChAT$^+$ neurons in the MSDB was achieved with a 50 mW, 473 nm laser (NovaPro Lasersytems, Germany) delivered through an optic fiber (200 μm, 0.22 numerical aperture) lowered into the MSDB. Light power measured at the tip of the fiber outside the brain was 3–4 mW. Pulses of blue light (5 ms) at 1 Hz (30 stimuli) were externally triggered using pClamp (Molecular Devices). Recordings were obtained with a Multiclamp 700B amplifier connected to the Digidata 1550 system. Data were acquired with pClamp 10 (Molecular Devices, Sunnyvale, CA, USA), digitized at 10 kHz, filtered at 3 kHz, and analyzed off-line with Clampfit 10.4 (Molecular Devices, Sunnyvale, CA, USA). Traces were high pass filtered (300 Hz) and events were detected using a threshold search function in clampfit. A burst was defined as a sequence of two or more action potentials occurring at ≥20 Hz (*Zucca et al., 2017*). A clampfit algorithm was used to detect bursts from the total events. The interval for burst detections ranged from 20 to 60 ms.

### Tissue preparation for immunohistochemistry

Tissue preparation and immunohistochemistry procedures were performed as previously described (*Modi et al., 2019*). Mice (aged P60–70) were anesthetized with i.p. injection of a mixture of tiletamine/zolazepam (zoletyl; 80 mg/kg body weight) and xylazine (rompun, 10 mg/kg body weight) and perfused transcardially with ice-cold oxygenated ACSF (pH 7.4) for 2 min (*Notter et al., 2014*). Brains were rapidly dissected and fixed for 48 h in 4% paraformaldehyde phosphate-buffered saline (PBS) solution (Santa Cruz, USA). After rinsing in PBS, brains were incubated with 30% (wt/vol) sucrose in PBS at 4 °C overnight, frozen with dry ice-cold isopentane and stored at −80 °C. Brains were embedded in the OCT compound (Leica, Germany) and sectioned by cryostat (Leica CM1850 UV, Germany).

### Histological verification of the cannula placements

Coronal brain sections (90-μm thick) were cut with a freezing microtome (Leica Microsystem, Germany). Serial slices were collected on gelatinized slides and stained with Cresyl Violet (Sigma-Aldrich, Italy). For the staining, slices were kept for 2 or 3 min in a 0.5% Cresyl Violet solution in distilled water. Immediately after, cleaning and dehydrating steps followed in this order: 1 min in distillated water for two times; 1 min in 50% EtOH, 1 min in 75% EtOH, 1 min in 100% EtOH, and 1 min in xylene substitute (Sigma-Aldrich, USA). Immediately after, slices were covered with cover slices and limonene (Sigma-Aldrich, Italy) as mounting medium. With the use of a stereomicroscope (Zeiss), the position of cannulas and injectors was verified and the most ventral point of the placement left by the injector during the administration was identified. Only animals with correct placements (within 350 μm from the CA2 border to CA1) were included in the statistical analysis. Illustration of coronal sections from animals were then represented for each pharmacology experiment.

### Immunohistochemistry

Free-floating coronal sections (60-μm thick) were rinsed in PBS 1×, permeabilized for 2 h at room temperature in blocking solution (1× DPBS; 0.3% Triton X-100; 5% normal donkey serum) and incubated overnight at 4 °C with combined primary antibodies in blocking solution. Antibody dilutions were as follows: anti-c-Fos (mouse, 1:150; Santa Cruz C-10, sc271243); anti-PCP4 (rabbit, 1:200; SIGMA, HPA005792); anti-ChAT (goat, 1:200; Millipore, AB144P); anti-PV (guinea pig, 1:1000; ImmunoStar, 24428). Following three 10 min washes in 1× DPBS/0.1% Triton X-100, slices were incubated with the appropriate donkey-raised secondary antibodies (1:500 dilution; Alexa Fluor, Thermo Fisher) for 2 h at room temperature, washed twice in 1× DPBS/0.1% Triton X-100 for 20 min, once in 1× DPBS/ DAPI for 20 min and then mounted with Aqua-Poly/Mount (Polysciences cat: 18606).

### Image acquisition

Z-stack images (16 optical sections, 0.4 μm step size) were recorded of all specimens using a spinning disk (X-Light V2, Crest Optics) microscope Olympus IX73 equipped with a LED light source (Spectra X light Engine, Lumencore, USA) and an Optimos camera (QImaging, Canada). Images were

acquired using a ×40 or ×20 objective with numerical aperture of 1.35 and 1.30, which had a pixel size of 108.3 × 108.3 and 106.2 × 106.2 nm$^2$, respectively. The Z-stacks were done with a motorized stage (HLD117, Prior Scientific, UK) controlled by MetaMorph software (Molecular Devices) and maximum intensity projections created for subsequent analyses. All imaging parameters were kept constant among samples and among experiments. The count of c-Fos$^+$ nuclei (% of c-Fos$^+$/DAPI$^+$ nuclei) was performed manually and the experimenter was not blinded to the treatment. A second experimenter repeated the analysis in blind and the results obtained were comparable. In hippocampi c-Fos$^+$ count was restricted to the CA2 area identified by the PCP4 marker. For each biological sample, the percentage of CA2 Fos$^+$ nuclei was calculated as a total from two hippocampal slices. In the MSDB, 2–3 coronal sections and a minimum of 3 fields/section were analyzed for each biological sample to identify an average of 79 ChAT$^+$ neurons/animal and calculate the percentage of c-Fos$^+$ cholinergic neurons.

### Analysis of axon density

Images were taken using a Zeiss LSM 700 confocal microscope with a 20× objective. For quantification of axonal densities, we analyzed Z-stacks of seven focal planes with a distance of 2 μm from each other. To reduce as much as possible the probability to count individual axons more than once we counted the number of axons in bins of 50 × 50 μm. We analyzed 10 bins per hippocampal area in one individual section. The sum of these 10 bins represents the axon density for one individual section. Four sections from each animal were analyzed. The final value of axon density for one hippocampal area from one mouse represents the mean of axon density from the four analyzed sections for this specific hippocampal area.

### Statistical analysis

Details of specific statistical designs and appropriate tests are described in each figure legend. Values are given as the mean ± SEM of $n$ experiments. No statistical methods were used to predetermine sample sizes, but our samples were in agreement with similar published studies. Significance of differences was assessed mainly by Student's paired or unpaired $t$-test and one-way ANOVA, as indicated. Wilcoxon or Mann–Whitney test was used for comparison of two groups and Tukey's test was used for more than two groups. Outliers were identified using ROUT method ($Q = 1\%$). Statistical differences were considered significant at $P < 0.05$. Statistical analysis was performed with GraphPad Prism 9.0 software (GraphPad, CA, USA).

## Acknowledgements

We thank M Mameli, R Pizzarelli, and M Rosato-Siri for comments on the manuscript. This work was supported by grants from the Veronesi's Foundation (Postdoctoral Fellowship 2016 and 2019 to MG), from Telethon (GGP 16083 to EC), from the European Union's Horizon 2020 Framework Programme for Research and Innovation under the Specific Grant Agreement No. 785,907 (Human Brain Project SGA2 to EC), from Sovena Foundation (Fellowship 2020 to DP), from Fondo Ordinario Enti (FOE D.M 865/2019) funds in the framework of a collaboration agreement between the Italian National Research Council and EBRI (2019–2021).

## Additional information

### Funding

| Funder | Grant reference number | Author |
| --- | --- | --- |
| Fondazione Telethon | GGP 16083 | Enrico Cherubini |
| Fondazione Umberto Veronesi | Fellowship 2016-2019 | Marilena Griguoli |
| Fondazione Sovena | Fellowship 2020 | Domenico Pimpinella |

| Funder | Grant reference number | Author |
| --- | --- | --- |
| Horizon 2020 - Research and Innovation Framework Programme | 785907 | Enrico Cherubini |

The funders had no role in study design, data collection and interpretation, or the decision to submit the work for publication.

## Author contributions

Domenico Pimpinella, Data curation, Formal analysis, Investigation, Methodology, Writing – review and editing; Valentina Mastrorilli, Data curation, Formal analysis, Investigation, Methodology; Corinna Giorgi, Data curation, Investigation, Methodology, Writing – review and editing; Silke Coemans, Salvatore Lecca, Arnaud L Lalive, Investigation; Hannah Ostermann, Elke C Fuchs, Formal analysis, Investigation; Hannah Monyer, Andrea Mele, Funding acquisition, Writing – review and editing; Enrico Cherubini, Funding acquisition, Writing – original draft, Writing – review and editing; Marilena Griguoli, Conceptualization, Data curation, Formal analysis, Methodology, Project administration, Supervision, Writing – original draft, Writing – review and editing

## Author ORCIDs

Salvatore Lecca http://orcid.org/0000-0001-5411-5485
Andrea Mele http://orcid.org/0000-0002-3155-5610
Marilena Griguoli http://orcid.org/0000-0003-4067-8927

## Ethics

All experiments were performed in accordance with the Italian Animal Welfare legislation (D.L. 26/2014) that were implemented by the European Committee Council Directive (2010/63 EEC) and were approved by local veterinary authorities, the EBRI ethical committee and the Italian Ministry of Health (565/PR18). All efforts were made to minimize animal suffering and to reduce the number of animals used.

## Decision letter and Author response

Decision letter https://doi.org/10.7554/eLife.65580.sa1
Author response https://doi.org/10.7554/eLife.65580.sa2

# Additional files

## Supplementary files

• Transparent reporting form

## Data availability

Data generated or analyzed during this study are included in the manuscript and supporting files. Source data files have been provided for Figures 2–9.

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
