## [Decision Letter]

**Acceptance summary:**

The authors describe results of experiments aimed at determining whether septal cholinergic inputs into hippocampal area CA2 play a role in social memory, a topic that is important and timely for a broad audience of neuroscientists interested in mechanisms of learning and memory and social behavior. The authors show that disruption of cholinergic neuron output impairs social memory as assessed by social novelty preference and that cholinergic input into the hippocampus, particularly in CA2, plays a substantial role in this finding.

**Decision letter after peer review:**

[Editors’ note: the authors submitted for reconsideration following the decision after peer review. What follows is the decision letter after the first round of review.]

Thank you for submitting your work entitled "Septal cholinergic input to CA2 hippocampal region controls social memory via nicotinic receptor-mediated disinhibition" for consideration by *eLife*. Your article has been reviewed by 3 peer reviewers, one of whom is a member of our Board of Reviewing Editors, and the evaluation has been overseen by a Senior Editor. The reviewers have opted to remain anonymous.

We are sorry to say that, after consultation with the reviewers, we have decided that your work will not be considered further for publication by *eLife*. That said, the reviewers were excited about the experimental question and the potential of the results, and so if you choose to expand the study to include new experiments according to their suggestions, you can submit it again as a new manuscript. In this case, please refer to the name of the editor, Laura Colgin, and your prior manuscript number, and every attempt will be made to recruit the same reviewers. Either way, we hope that you will find the reviewer comments useful in revising your manuscript for future submission(s).

*Reviewer #1:*

Pimpinella, et al. describe results of experiments aimed at determining the role of septal cholinergic inputs into hippocampal area CA2 in social memory. On one level, the authors show compelling evidence, using two different methods, that disruption of cholinergic neuron activity or output impairs social memory as assessed by social novelty preference. However, the primary weakness is that the authors have not yet proven a specific role for CA2 or CA2 interneurons in the behavior; area CA3 seems to have just as much MS axonal coverage as does area CA2, and the local drug infusions in the hippocampus would not appear to be selective to CA2. At the least, the authors should analyze the Fos staining in CA1 and CA3, which appear to have much less (in the case of CA1, or similar in the case of CA3) MS axonal coverage. In addition, some key control experiments and details are missing. The experiments address important aspects of both social behavior and cholinergic signaling, however more experiments in vitro and in vivo, performed in CA1 and CA3 really should be performed, or else the conclusions narrowly implicating CA2 will need to be modified to include the other CA regions.

The experiments and analyses appear to be performed to a high standard, and the manuscript is well written and the data nicely presented. The main findings, that inhibition of MSDB cholinergic neurons impairs social memory, are an important contribution to the field. Other conclusions, such a specific role for ChAT+ neuron influence on CA2 pyramidal neurons, through CA2 interneurons, impacting social memory, are less well supported by the data presented. Although generally supportive of the authors' hypothesis, little data was presented to test whether the behavioral effects of ACh release inhibition are due to the influence on hippocampal areas with similar (CA3) or less (CA1) MS axonal input, as their data would also support similar conclusions involving those areas (Fos staining blocked by CNO and electrophysiological effects in vivo and in vitro, for example, were not tested for other areas). Therefore, a number of additional experiments/analyses, listed below, would be required to make the case.

1. The whole justification for studying CA2 without comparing it to CA1 and CA3 rests on the reported role for CA2 in social memory and not say, a distinctive input from the MSDB. However, the authors' Figure 1D does seem to show some differences in ChAT axon patterns in the different areas. The authors could make a stronger justification for the focus on CA2 if they present some type of quantification of the terminal labeling in CA1/CA2/CA3 stratum pyramidale vs. radiatum vs. lucidum, for example.

2. The authors found that social interaction elicited a strong increase in c-Fos-positive cells in the MS and in CA2, which was prevented in DREADD silenced mice (Figure 3E). CA2 typically has the least amount of Fos stain of all the hippocampal subfields, and it looks from the images presented that there is a strong increase in Fos in what may be CA1, which also looked to be inhibited by CNO. As MS projects strongly to CA3, and to a lesser extent, CA1, the authors will need to present the Fos data from CA3 and CA1 as similar increases and disruptions by MS silencing would weaken the idea that CA2 is substantially different in MS influence on hippocampal neuronal activity (this not necessarily problematic, as CA1 could be activated by CA2, but it would need to be discussed in this context if so).

3. Also, are the Fos+ cells in CA2 the ones also positive for PCP4? This is unclear from the images (and almost looks like they are not). If not, did the authors also co-stain for inhibitory markers? (this would be important given that the authors are proposing a role for inhibitory neurons, and an increase in Fos in them would not support the authors' proposal that MS input to interneurons is decreasing their firing).

4. The authors should consider determining whether MSDB Fos induction (Figure 3 suppl 2) is specific for social interaction. Does novel object and/or novel environment similarly increase Fos staining? Same could be asked regarding CA2 Fos.

5. The synaptic effects of MSDB axonal silencing and nicotine in vitro (Figure 6 and Figure 4 suppl) in CA2 are very interesting, but the results are lacking context in relation to the axonal coverage shown in Figure 1 (again, comparing CA1 and/or CA3). One would expect a dramatically smaller effect in CA1 and a similar effect in CA3. Additionally, the conclusions could be strengthened by experiments using excitatory DREADDs or optogenetic stimulation.

6. Likewise, the optogenetic activation of cholinergic neurons in MSDB increases the firing of CA2 principal cells in vivo is an exciting result, but data from CA3 and/or CA1 would reveal whether there is anything fundamentally different between the areas (such as what might be expected if the ACh receptor distribution is different, like chrm3, and perhaps chrna4. See http://mouse.brain-map.org/gene/show/12456)

7. The conclusions based on the data in Figure 7 would be stronger if the authors would have included light-only controls, ideally in Chat-cre-negative- littermates. Although the light in MS is unlikely to be directly affecting CA2 neurons, heat generated in the MS but could be activating ChAT- neg neurons projecting to the hippocampus.

Further comments

1. In some places, the authors use 'impairs social novelty discrimination", which is a good description of what was actually measured. This should be used throughout the manuscript instead of 'memory', which is an interpretation (better brought up in the Discussion).

2. Methods: please expand on the methods used to count Fos+ nuclei (software for counting or manually counted? If manually, was the experimenter blinded to treatment?)

3. Methods: please include a statement justifying use of male mice only.

4. Results heading (line 72): "ACh released from cholinergic neurons in the MSDB is required for social novelty" should be '… social novelty discrimination'.

5. Legend in S2. (C and D) (and throughout) are not really a scatter plots.

6. Fos images are labeled with 'social behavior'. More accurate would be 'social interaction'.

7. The axis labels for the Fos staining in Figure 3 suppl 2 (D) is unclear. In the methods, the authors state "for each biological sample, the percentage of CA2 Fos+ nuclei was calculated as a total from two hippocampal slices and normalized to that obtained in the HCC condition" but the axis is labeled "% of fos+ cholinergic neurons". Do the authors mean "% of cholinergic neurons that are Fos+"? Similarly for Figure 3 F "normalized % of Fos+ nuclei". Should be "% of nuclei that are Fos+ (normalized to HCC)".

8. The graphs in Supp 3-2 'D' and Figure 3F are low resolution.

9. Please label CA1/CA2/CA3 on images.

10. Wording on lines 183-4: "In slices obtained from naive mice not expressing hM4, CNO did not affect the frequency and the amplitude of both sIPSCs (Figure 4-supplement 2C) and sEPSCs (Figure 4-supplement 2D)." should read: ".. CNO had no effect on either frequency or amplitude of sIPSCs or sEPSCs." Or something similar.

11. Methods, line 494: spelling should be Xylazine, not Xilazine (unless brand name? please confirm).

12. Methods, line 432: please clarify or be consistent with usage of 'spontaneous' synaptic currents vs. 'miniature' currents. (stated: "Miniature GABAA-mediated inhibitory postsynaptic currents…", but abbreviate with sIPSCs)". If TTX was used in both cases, why the difference in nomenclature? (should be mEPSC and mIPSC in that case, no?).

13. Methods, line 498: "frost" should be "frozen".

14. References: should cite Raam, et al., 2017, showing role for CA2 in social discrimination.

*Reviewer #2:*

In this manuscript, the authors investigate the role of acetylcholine in social memory and sociability. They combine pharmacological experiments with chemogenetics, in-vivo experiments and different behavior paradigms. In general, the proposed question of how a common neuromodulator such as ACh regulate social memory is of general interest. The authors find that social novelty recognition it is mediated by ACh release from Medial Septum, which specifically activates nicotinic receptors in local interneurons, what contributes to disinhibition of CA2 pyramidal neurons. The main caveat of the study is the lack of temporal resolution for most of the points regarding the mechanisms by which ACh controls different aspects of social memory. The nature of these experiments assumes social memory as a whole part, while it has been shown that social (and in general any type of memory) it is further subdivided in different processes (encoding, consolidation, recall…) which in turn involve different brain mechanisms. Yet, I think that in general the authors made a good effort combining different techniques to dissect the present mechanisms that could potentially relate ACh with social memory. Some specific points that could strengthen the main conclusions are detailed below.

1. In Supplemental Figure 1 (panel C), the authors show that after applying TeNT the frequency of spiking is not affected, however, there is a clear trend of a lower frequency with TeNT conditions, suggesting that there might be a masked effect due to low N.

2. In general, it could be nice to clarify in every part how long after infusion of different drugs were the animals tested, I couldn't find it for example for the TeNT experiments.

3. The cFos experiments are nice but lack the temporal resolution to correlate this activity dependent marker with any specifics of the task, i.e., social interactions/memory. It cannot be rule out that the cFos activity is free movement of the animal in any task (since there are no controls with any other behavioral paradigm). For this and in general for all group of experiments, it would be helpful to compare it with the responses obtained in the object recognition test (novel/familiar).

4. In line 358 the authors claim that "local release of ACh in the CA2 is sufficient for social memory's encoding". I think that what it can be proved with their experiments is that local release of ACh in the CA2 is necessary for social memory's encoding. In addition, the title of this part states "Local release of ACh in the CA2 hippocampal region is necessary for social novelty", shouldn't it be "for social novelty encoding/detection/recognition"?

5. In the main text, it is stated that n=14/n=13 animals were used for optogenetics groups, while in the figure 4 legend it is n=12/n=10. Were some animals excluded maybe due to virus or canula mistargeting?

6. Could the authors provide any insight with immunostaining (or even just speculate/discuss it), on which type of interneurons might be being involved in disinhibiting CA2 cells?

*Reviewer #3:*

The current study by Pimpinella et al. investigates an important and timely question regarding the contribution of the neuromodulator acetylcholine to social memory. By conducting behavioral experiments with chemogenetic, pharmacological, and optogenetic manipulations of cholinergic neurons in the medial septum or cholinergic release sites in the hippocampal CA2 region in mice, the authors demonstrate that cholinergic signaling is critically important for social memory and also for social novelty-related neuronal activity in CA2. Cholinergic neuromodulation in the hippocampus has long been implicated in novelty encoding but the dependence of social memory on cholinergic modulation has not been shown so far. The authors further show that cholinergic modulation of social memory depends on nicotinic acetylcholine receptors as opposed to muscarinic receptors, an important step towards deciphering the mechanism of cholinergic action contributing to social memory encoding. The authors claim that they have identified the mechanism of cholinergic action underlying social memory formation as disinhibition of CA2 principal neurons via activation of nicotinic acetylcholine receptors on CA2 interneurons. However, this claim is currently only poorly supported by their experimental data and the authors do not rule out alternative hypotheses that may account for the observed effects. The authors further claim that the inhibition of cholinergic modulation in the CA2 region specifically affects social memory but no other types of memory such as spatial memory. However, they do not present convincing evidence that spatial memory is spared and that inhibition of cholinergic signaling is constrained to the CA2 region. In summary, the manuscript presents very interesting data on the importance of cholinergic signaling for social memory. However, the manuscript in its current form lacks critical evidence supporting the authors' claims regarding the proposed mechanism of cholinergic modulation of social memory.

1. With respect to the TeNT experiments: The authors convincingly demonstrate that cholinergic MSDB neurons are important for social memory. However, they then further claim in lines 102-107 that the memory effect is specific for social memory based on comparing the effect on social memory with effects on novel object recognition. I am not convinced that this is the correct control experiment because social memory is hippocampus-dependent, while novel object recognition is not (or at least there is no strong evidence for novel object recognition being hippocampus-dependent). It is odd that the authors chose the NOR test as opposed to a hippocampus-dependent test such as the novel object location task. If the authors want to make the point that cholinergic signaling is specifically important for social memory, they would need to show that other forms of hippocampal-dependent memory such as spatial memory is unaffected or less affected by the same manipulations. Given the current data, it seems that hippocampal acetylcholine release from MSDB cholinergic projection neurons is important for hippocampus-dependent novelty tasks including-but not necessarily specific-to CA2-dependent social memory.

2. Lines 89-91, Figure 2—figure supplement 1C: "No changes in the (…) spontaneous firing frequency between eYFP and TeNT expressing neurons were observed." The authors only compared n = 5 TeNT cells with n = 6 eYFP cells using a non-parametric test that is underpowered at a sample size of 5. Nevertheless, they observe a more than threefold reduction in firing rate with a p-value of p = 0.08. This is very close to significance despite the low sample size and the test being underpowered. It is very likely that the authors would find a significant difference if they increase the sample size to n > 10. I am concerned that TeNT does not only affect synaptic release of vesicles but also basic firing properties of the cells. The authors may argue that, even if that is the case, the main conclusion of the paper may not change because the overall effect of TeNT expression is an inhibition of cholinergic activity. However, it should be discussed that TeNT may not be specifically affecting only synaptic acetylcholine release.

3. With respect to systemic CNO experiments: It has been shown by Gomez et al. (2017) that CNO is very likely not the active component responsible for DREADD-effects but that CNO is instead converted to clozapine which then acts on DREADD receptors. The authors cite this paper and do the correct control experiments by comparing their results with CNO-injected eYFP mice. However, I am missing a more detailed discussion when introducing the method (line 120) and a short discussion about potential off-target effects that has been previously shown for the CNO concentrations used in the current study (10 µM for in vitro experiments and 100 µM or 3 mg/kg for in vivo experiments).

4. Lines 145-147: "(…) social behavior elicited strong increase in c-Fos-positive cells (in the MSDB)". The authors do not provide evidence that c-Fos induction is caused specifically by social novelty as opposed to general novelty or exposure to the three-chamber test apparatus. The authors mention in the Discussion section that "(…) to rule out the contribution of c-Fos expression induced by a novel context, the animals were exposed to the three-chamber apparatus 24 hours before the social test" (lines 269-271). However, a single exposure to the test chamber one day before the test does very likely not reduce c-Fos expression on the next day. If the authors want to make that claim, they need to provide data showing that a single exposure to the same environment on the previous day significantly reduces c-Fos activation. As it stands, the current data do not show that social memory is related to c-Fos activation in the MSDB or hippocampus in addition to the c-Fos activation generally observed when taking mice out of their home cage and placing them in a test chamber.

5. With respect to experiments using local injections of CNO or nicotine into the CA2 region, how can the authors be sure that CNO and nicotine (both injected at relatively high concentrations) do not diffuse to the surrounding areas CA1 and CA3? A caveat of those experiments is that the inhibition of cholinergic signaling may not be constrained to the CA2 region. Observed effects of cholinergic inhibition may therefore be completely or partially caused by blocking cholinergic action in surrounding areas CA1 and/or CA3. This point becomes even more important in the light of the results of the local CNO injection, which-contrary to what the authors argue in the manuscript-did not show a significant difference between eYFP and hM4 animals.

Related to this point, the authors claim that "local release of ACh in the CA2 is sufficient for social memory's encoding" (line 168). However, this claim is not supported by their data. The data shown in Figure 4C clearly show that there is NO significant difference in the difference score between eYFP mice and hM4 mice (right panel). It would be wrong to draw that conclusion simply because there is a significant difference between Familiar and Novel in the eYFP group but not the hM4 group (see Nieuwenhuis et al., 2011). The correct statistical comparison is the one shown in the right panel (as acknowledged by the authors themselves earlier in the manuscript when introducing the difference score).

A similar consideration applies to data presented in Figure 4 —figure supplement 2. Since CNO can have off-target effects, the authors should statistically compare effects of CNO in hM4 mice to effects of CNO in naïve mice.

6. Regarding experiments on nicotinic control of neuronal activity in CA2: The authors claim that the mechanism of nicotinic receptor-dependent social memory is disinhibition of principal neurons in area CA2 via nicotinic activation of interneuron-selective interneurons. While this is an intriguing model consistent with their data, the authors do not provide convincing evidence for such a model or mechanism. In particular, recordings of interneurons are missing to support the author's conclusion and alternative hypotheses have not been addressed.

7. Previous studies (Malezieux et al., 2020 and Dannenberg et al., 2015) have shown that brain states associated with high cholinergic activity and theta oscillations result in reduced firing of CA3 principal neurons via activation of interneurons. If the net effect of acetylcholine release in CA2 is activation of principal neurons via disinhibition as opposed to inhibition of principal neurons as shown in CA3, that would be a very interesting finding. However, the authors should discuss the effect of tiletamine/xylazine anesthesia in their experiments. Moreover, the authors could address alternative explanations. For example, cholinergic stimulation could result in a general increase of network activity resulting in higher firing rates in both interneurons and principal neurons. Data on interneurons would help distinguish between those hypotheses.

8. I couldn't find details on optogenetics experiments in the Methods section (e.g., light power, wavelength, viral construct).

References:

Dannenberg, H., Young, K., and Hasselmo, M. (2017). Modulation of Hippocampal Circuits by Muscarinic and Nicotinic Receptors. Frontiers in Neural Circuits, 11, 102. https://doi.org/10.3389/fncir.2017.00102

Haam, J., and Yakel, J. L. (2017). Cholinergic modulation of the hippocampal region and memory function. Journal of Neurochemistry, 142 Suppl 2, 111-121. https://doi.org/10.1111/jnc.14052

Nieuwenhuis, S., Forstmann, B. U., and Wagenmakers, E.-J. (2011). Erroneous analyses of interactions in neuroscience: A problem of significance. Nature Neuroscience, 14(9), 1105-1107. https://doi.org/10.1038/nn.2886

McQuiston, A. R. (2014). Acetylcholine release and inhibitory interneuron activity in hippocampal CA1. Frontiers in Synaptic Neuroscience, 6. https://doi.org/10.3389/fnsyn.2014.00020

Malezieux, M., Kees, A. L., and Mulle, C. (2020). Theta Oscillations Coincide with Sustained Hyperpolarization in CA3 Pyramidal Cells, Underlying Decreased Firing. Cell Reports, 32(1). https://doi.org/10.1016/j.celrep.2020.107868

Dannenberg, H., Pabst, M., Braganza, O., Schoch, S., Niediek, J., Bayraktar, M., Mormann, F., and Beck, H. (2015). Synergy of direct and indirect cholinergic septo-hippocampal pathways coordinates firing in hippocampal networks. The Journal of Neuroscience, 35(22), 8394-8410. https://doi.org/10.1523/JNEUROSCI.4460-14.2015

Further comments:

1. Title and elsewhere (e.g., line 69): "Septal cholinergic input (…) controls social memory (…)." The wording "controls social memory" is very vague. Can the authors describe the major finding of the manuscript more precisely?

2. Lines 29-30 and elsewhere: Patch clamp recordings in hippocampal slices are usually referred to as "in vitro" experiments as opposed to "ex vivo". Is there any specific reason why the authors have chosen "ex vivo" as opposed to "in vitro"?

3. Lines 59: The authors could cite more recent reviews of the cholinergic system in addition to the reference to Teles-Grilo Ruivo and Mellor (2013), for example Dannenberg et al. (2017) and Haam and Yakel (2017).

4. Figure 2: I find the way the symbols and colors are used confusing. For example, the circle symbol is used for Animal, Object, Novel, and Familiar. It is further confusing that in E, orange and blue colors are used to indicate Novel and Familiar, but in B, D, and F, orange and blue colors are used to indicate eYFP and TeNT.

5. Line 173: "isolated pharmacologically". How?

[Editors’ note: further revisions were suggested prior to acceptance, as described below.]

Thank you for resubmitting your work entitled "Septal cholinergic input to CA2 hippocampal region controls social novelty discrimination *via* nicotinic receptor-mediated disinhibition" for further consideration by *eLife*. Your revised article has been evaluated by Laura Colgin (Senior Editor) and a Reviewing Editor.

Essential revisions:

The authors describe results of experiments aimed at determining whether septal cholinergic inputs into hippocampal area CA2 play a role in social memory, a topic that is important and timely for a broad audience of neuroscientists interested in mechanisms of learning and memory and social behavior. The authors show that disruption of cholinergic neuron output impairs social memory as assessed by social novelty preference and that cholinergic input into the hippocampus, particularly in CA2, are playing substantial role in this finding. The revised manuscript is much improved from the first submission and the authors present some exciting new data that contrasts CA2 with the other hippocampal subregions. Nevertheless, there are still several issues that should be addressed in the text and relating to the figures before it is acceptable for publication.

*Reviewer #1:*

This study is an important contribution to the field and is likely of interest to a broad readership of neuroscientists. The revised manuscript is much improved from the first submission and presents some exciting new data. Although the authors addressed most of the reviewer concerns, there are still a number of issues that should be addressed in the text before it is acceptable for publication. These, along with some minor issues, are listed below.

1. The authors now include data from Novel Object Recognition and Object Location tests (Figure 3E, F), however they still did not show whether investigation of a Novel Object induces Fos in a way similar to that induced by the social stimulus. The data now included showing Fos in response to exploration of an empty arena and home cage controls are important additions, but the possibility that an increase in Fos staining in CA2 may not be selective for a social stimulus should be clearly acknowledged and discussed since a similar response to a novel object has not been ruled out. This is an important point given that place field remapping occurs with novel objects to a similar degree as that occurring with a social stimulus (Alexander, 2016).

2. The authors should address the locations of the cannulas, several of which look like they were in CA1. In this case, they should acknowledge in the discussion the limitations of the method.

3. It was interesting that most of the Fos+ neurons in CA2 were likely interneurons, and so it is somewhat surprising that the authors still do not show recordings from interneurons. The findings do not appear to be overstated ( "…provides insight into the mechanism" in the abstract), but some discussion stating the limitations of the study is warranted. These recordings would be critical for supporting the authors model of cholinergic regulation of CA2 output, and so the authors are encouraged to submit those recordings as an *eLife* "Research Advance".

4. The graphic is confusing in light of (1) the new data showing the increase in Fos in the interneurons, (2) no recordings from interneurons, and (3) there is some discrepancy with the observation of increased firing of CA2 pyramidal cells in response to opto-stimulation. I suggest deleting the graphic unless it can be revised to better reflect the findings.

*Reviewer #2:*

In the revised version of the manuscript, the authors have added new experimental support to their previous point, which substantially helped to strengthen the points made. Increasing the n numbers for several measures also helped to make sure the conclusions are solid. I am more convinced now about the mechanism that they proposed, stating that ACh, via nicotinic receptors, is necessary for encoding social novelty. However, although is interesting that they provide several possible models at the end, I still regret the lack of more specific data in relation to the type of interneurons. Overall, I think the authors made a decent job tackling most of the points raised in the previous review round and the information provided in this manuscript will be altogether of interest for the community.

Figure 1

– Units in figure 1 for axon density? Should it be "fluorescence density (AU, X20)"?

– 676 line reads: "stacks of 7 focal planes with a distance of 2?m" should it be "stacks of 7 focal planes with a distance of 2mm"

– No background subtraction was used I assume?

Figure 2

– This figure, still compares "stimulus" versus "non-stimulus" condition, without assessing whether the responses cFos+ are exclusive for an animal (social) or could be anything, object, spatial trajectory, foraging…

Figure 7

– Scales missing in A and C traces (or are they all the same one?), please clarify.

Figure 8

– Scales missing in A and C traces.

Figure 9

– Duration of the bursts would also be informative.

– Number of bursts would also be informative.

Figure 5—figure supplement 2

– Temporal scale missing in traces showed in A.

– Scales missing in traces showed in E.

– Y labels in I and J are confusing (shown in "%"), sIPSCs frequency is shown in Hz (in the above plots), so perhaps the authors meant difference between conditions or ratio compare to control in these plots?

Figure 7—figure supplement 1

– Temporal scale missing in trace showed in A.

Figure 7—figure supplement 2

– Temporal scale missing in trace showed in A.

Figure 9—figure supplement 1

– Temporal scale missing in trace showed in B.

Figure 9—figure supplement 2

– It is interesting that the spiking frequency of CA3 neurons is not affected at all by optical stimulation of MSDB, despite the frequency of sEPSCs being affected by nicotine.

– Temporal scale missing in trace showed in B.

*Reviewer #3:*

The revised manuscript entitled "Septal cholinergic input to CA2 hippocampal region controls social novelty discrimination via nicotinic receptor-mediated disinhibition" by Pimpinella et al. now presents additional experimental data and analyses that address previous concerns raised by reviewers. In particular, the authors now demonstrate that local delivery of CNO to the CA2 region of hM4-expressing mice impairs social novelty detection. Furthermore, the authors added in vitro experimental data that demonstrate an increase in spontaneous EPSCs in the presence of nicotine in CA2 but not in CA1 and CA3. In fact, a decrease in spontaneous EPSCs is observed in CA3. These new data provide evidence that nicotinic cholinergic receptors differentially affect network activity in hippocampal subregions and support the authors' hypothesis that cholinergic disinhibition of CA2 via nicotinic receptors is important for social novelty discrimination. Furthermore, the authors now report experimental results that demonstrate that social interaction increases cFos expression in cholinergic neurons in the medial septal complex further highlighting a significant activation and contribution of the septal cholinergic system during social interaction. I have few remaining comments, mostly addressing statistical analyses and the clarity of data presentation.

1. Regarding major point 2 (TeNT experiments): Do the authors now interpret their findings as an effect of TeNT on intrinsic firing properties? To more clearly distinguish between the alternative and null hypotheses that TeNT has or has not an effect on intrinsic firing properties, I suggest a statistical analysis using Bayes factor (Keysers, Christian, Valeria Gazzola, and Eric-Jan Wagenmakers. "Using Bayes Factor Hypothesis Testing in Neuroscience to Establish Evidence of Absence." Nature Neuroscience 23, no. 7 (July 2020): 788-99. https://doi.org/10.1038/s41593-020-0660-4.; Dienes, Zoltan. "Using Bayes to Get the Most out of Non-Significant Results." Frontiers in Psychology 5 (2014): 781. https://doi.org/10.3389/fpsyg.2014.00781). The Bayes factor will indicate whether (a) the data support the null hypothesis or (b) the data support the alternative hypothesis, or (c) the sample size is too low to draw a conclusion.

2. Why does the eYFP virus has a different serotype (2/9) than the TeNT carrying AAV (DJ serotype)? The possibility that diverse tropism of eYFP carrying AAV vs TeNT carrying AAV may target different subtypes of cholinergic neurons – as stated by the authors in the rebuttal letter – should also be discussed in the manuscript itself.

3. To Figure 3C and Figure 4C: It is not immediately clear from looking at the figure if Novel and Familiar refers to an object or to an animal. I suggest to replace the circles with symbols that make this immediately clear to the reader (e.g., a drawing of a mouse in Figure 3C and drawing of an object in Figure 3A).

4. Figure 4 —figure supplement 5: In my opinion, this figure should be part of main Figure 4 because it is an important control experiment and shows that chemogenetic inactivation of cholinergic neurons in the medial septal complex also affects spatial memory in addition to social memory, though the effect size appears smaller for effects on spatial memory compared to the larger effects on social memory. But I leave the final decision to the authors.

5. Line 211: "necessary" instead of "sufficient"?

---

## [Author Response]

[Editors’ note: The authors appealed the original decision. What follows is the authors’ response to the first round of review.]

Reviewer #1:Pimpinella, et al. describe results of experiments aimed at determining the role of septal cholinergic inputs into hippocampal area CA2 in social memory. On one level, the authors show compelling evidence, using two different methods, that disruption of cholinergic neuron activity or output impairs social memory as assessed by social novelty preference. However, the primary weakness is that the authors have not yet proven a specific role for CA2 or CA2 interneurons in the behavior; area CA3 seems to have just as much MS axonal coverage as does area CA2, and the local drug infusions in the hippocampus would not appear to be selective to CA2. At the least, the authors should analyze the Fos staining in CA1 and CA3, which appear to have much less (in the case of CA1, or similar in the case of CA3) MS axonal coverage. In addition, some key control experiments and details are missing. The experiments address important aspects of both social behavior and cholinergic signaling, however more experiments in vitro and in vivo, performed in CA1 and CA3 really should be performed, or else the conclusions narrowly implicating CA2 will need to be modified to include the other CA regions.The experiments and analyses appear to be performed to a high standard, and the manuscript is well written and the data nicely presented. The main findings, that inhibition of MSDB cholinergic neurons impairs social memory, are an important contribution to the field. Other conclusions, such a specific role for ChAT+ neuron influence on CA2 pyramidal neurons, through CA2 interneurons, impacting social memory, are less well supported by the data presented. Although generally supportive of the authors' hypothesis, little data was presented to test whether the behavioral effects of ACh release inhibition are due to the influence on hippocampal areas with similar (CA3) or less (CA1) MS axonal input, as their data would also support similar conclusions involving those areas (Fos staining blocked by CNO and electrophysiological effects in vivo and in vitro, for example, were not tested for other areas). Therefore, a number of additional experiments/analyses, listed below, would be required to make the case.1. The whole justification for studying CA2 without comparing it to CA1 and CA3 rests on the reported role for CA2 in social memory and not say, a distinctive input from the MSDB. However, the authors' Figure 1D does seem to show some differences in ChAT axon patterns in the different areas. The authors could make a stronger justification for the focus on CA2 if they present some type of quantification of the terminal labeling in CA1/CA2/CA3 stratum pyramidale vs. radiatum vs. lucidum, for example.

We followed the Reviewer’s suggestion and a quantification of cholinergic terminals labeling in CA1/CA2/CA3 was included in the New figure 1 (Panel E).

2. The authors found that social interaction elicited a strong increase in c-Fos-positive cells in the MS and in CA2, which was prevented in DREADD silenced mice (Figure 3E). CA2 typically has the least amount of Fos stain of all the hippocampal subfields, and it looks from the images presented that there is a strong increase in Fos in what may be CA1, which also looked to be inhibited by CNO. As MS projects strongly to CA3, and to a lesser extent, CA1, the authors will need to present the Fos data from CA3 and CA1 as similar increases and disruptions by MS silencing would weaken the idea that CA2 is substantially different in MS influence on hippocampal neuronal activity (this not necessarily problematic, as CA1 could be activated by CA2, but it would need to be discussed in this context if so).

A quantification of c-Fos-positive nuclei in CA3 and CA1 was performed in both eYFP and hM4expressing mice that were subjected to social behavior and that received 30 minutes prior to the social novelty task a systemic injection of CNO. Home-caged animals, not exposed to behavioral tasks, were used as controls. c-Fos quantification in CA3 and CA1 was performed on the same sections analyzed for CA2 (CA2 results in Figure 3E of the previous manuscripts’ version are now shown in Figure 4E-F). No significant changes in the % of nuclei expressing c-Fos were detected among the three experimental groups. These results suggest that under these conditions there is not a selective activation of c-Fos in response to social interaction in CA3 and CA1 (New Figure 4figure supplement 4).

3. Also, are the Fos+ cells in CA2 the ones also positive for PCP4? This is unclear from the images (and almost looks like they are not). If not, did the authors also co-stain for inhibitory markers? (this would be important given that the authors are proposing a role for inhibitory neurons, and an increase in Fos in them would not support the authors' proposal that MS input to interneurons is decreasing their firing).

We thank the Reviewer for this suggestion and we proceeded to analyze whether c-Fos positive cells are also positive for the PCP4 marker. Indeed, in our sections (n=3 animals, average n=8 c-Fos^+^ nuclei/animal) 94.1 ± 3 % of c-Fos expressing cells were PCP4^-^ (Figure 4—figure supplement 2A-B), confirming the Reviewer’s observation. Previous studies showed that the PCP4 marker overlaps with different CA2 markers expressed by principal neurons (Kohara et al., 2014). It was shown that PCP4-expressing neurons are putative pyramidal cells as they express CAMKII, a glutamatergic marker (Kohara et al., 2014). However, we cannot exclude the possibility that there is a subpopulation of PCP4^-^ principal cells within the CA2 boundaries. To investigate whether PCP4^-^ neurons were GABAergic interneurons, we performed double immunostaining for parvalbumin (PV) and c-Fos. The use of this marker was justified by the fact that PV^+^ neurons frequently have large pyramid shaped or fusiform somas that are typically localized within or in immediate vicinity to *stratum pyramidale* (Pelkey et al., 2017). Furthermore, PV^+^ neurons contributing to cortical disinhibitory circuits (Pi et al., 2013; Pfeffer et al., 2013). Our analysis revealed that the majority of c-Fos^+^ cells were PV^-^ (91.5 ± 4.5 %; n=3 animals; average n=8 c-Fos^+^ nuclei/animal) (New Figure 4—figure supplement 2A-C). To better identify which subtypes of GABAergic interneurons maybe involved in social interactions is indeed a very interesting question that would need, however, a complete new study and hence cannot be addressed in the present work.

4. The authors should consider determining whether MSDB Fos induction (Figure 3 suppl 2) is specific for social interaction. Does novel object and/or novel environment similarly increase Fos staining? Same could be asked regarding CA2 Fos.

We performed the requested experiments to exclude the possibility that c-Fos expression in the MSDB was induced by locomotor activity and environment exploration (a point raided also by Reviewer 2 in point 3). To this purpose mice (n = 4) were habituated to the three chamber arena (empty, EA) for 10 minutes (day 1) and again for 10 minutes after 24 hours (day 2). In this way, the animals underwent to the same handling and habituation phases as those exposed to social stimuli during the three chamber test (social interaction, SI). After one hour, we sacrified the animals. c-Fos immunolabeling revealed a certain degree of c-Fos expression after EA exploration as compared to HCC controls (New Figure 2, panels 2D-F). Nevertheless this was significantly lower than what observed in the SI condition (New Figure 2, panels 2G-H). Interestingly, we did not detect any ChAT^+^ neuron expressing c-Fos in the EA group, indicating that ChAT^+^ neurons selectively express c-Fos in response to social stimuli (New Figure 2, panels A-C, H).

Previous studies showed that a novel environment activates early gene transcription in the CA2 neurons (VanElzakker et al., 2008; Wintzer et al., 2014) and this activation is comparable to that induced by social stimuli (Alexander et al., 2016). In our case, the animals were habituated to the novel environment and the only discriminating factor between EA and SI conditions was the social stimuli. We performed c-Fos immunolabeling in EA group and we did not find a difference in the % of c-Fos^+^ nuclei in CA2 as compared to HCC (New Figure 4-supplementary 3).

5. The synaptic effects of MSDB axonal silencing and nicotine in vitro (Figure 6 and Figure 4 suppl) in CA2 are very interesting, but the results are lacking context in relation to the axonal coverage shown in Figure 1 (again, comparing CA1 and/or CA3). One would expect a dramatically smaller effect in CA1 and a similar effect in CA3. Additionally, the conclusions could be strengthened by experiments using excitatory DREADDs or optogenetic stimulation.

We collected new data regarding the effects of nicotine on synaptic transmission recorded from both CA3 and CA1 pyramidal cells. Contrary to what we previously observed in the CA2 region, where nicotine increased the frequency of spontaneous excitatory postsynaptic currents (sEPSCs), we found a reduced frequency of sEPSCs in both CA3 and CA1 regions (New Figure 7—figure supplement 2). As already reported (Rosato-Siri et al., 2006; Tang et al., 2011; Hajós et al., 2005), the reduced frequency of sEPSCs in both CA3 and CA1 regions may be due to a nAChRs-dependent increase of GABAergic tone with consequential reduction of glutamatergic transmission.

Following Reviewer’s suggestion we injected the MSDB of ChAT-Cre mice with an AAV carrying the floxed sequence of hM3 excitatory DREADD. This allowed to the effects of endogenously released ACh on sEPSCs in CA3, CA2 and CA1 principal neurons before and during bath application of CNO. According to the effect observed in the presence of nicotine, CNO increased the frequency but not the amplitude of sEPSCs in CA2 principal neurons. In CA3, CNO mimicked the effect of nicotine by decreasing the frequency of sEPSCs. CNO also induced a significant decrease in sEPSCs amplitude, not observed during nicotine application. This discrepancy could be explained by different affinity and/or desensitization of nAChR subtypes when bound by ACh or nicotine (Albuquerque et al., 2009). In CA1, CNO did not change either the sESPCs frequency or the amplitude, possibly because of a reduced coverage of hM4-expressing fibers and hence an insufficient ACh release in response to hM3 activation (New Figure 8).

6. Likewise, the optogenetic activation of cholinergic neurons in MSDB increases the firing of CA2 principal cells in vivo is an exciting result, but data from CA3 and/or CA1 would reveal whether there is anything fundamentally different between the areas (such as what might be expected if the ACh receptor distribution is different, like chrm3, and perhaps chrna4. See http://mouse.brain-map.org/gene/show/12456)

Juxtacellular recordings combined with photo stimulation of ChAT^+^ neurons expressing ChR2 in the MSDB from both CA3 and CA1 regions were performed. No changes in the firing frequency or intraburst frequency from both CA3 and CA1 neurons could be observed during light stimulation. This may be related to the concomitant activation of mAChRs, which would probably counteract the depressing effect of nAChRs on glutamatergic transmission. These results were included in a new supplementary figure (Figure 9—figure supplement 2).

7. The conclusions based on the data in Figure 7 would be stronger if the authors would have included light-only controls, ideally in Chat-cre-negative- littermates. Although the light in MS is unlikely to be directly affecting CA2 neurons, heat generated in the MS but could be activating ChAT- neg neurons projecting to the hippocampus.

We agree with Reviewer that the experiment she/he suggested is an important control. Thus, juxtacellular recordings combined with light delivery in the MSDB were performed from the CA2 region of ChAT-Cre mice lacking ChR2 expression. No change in the frequency of spontaneous firing was observed in recorded neurons. These data were added to Figure 9—figure supplement 1.

Further comments:1. In some places, the authors use 'impairs social novelty discrimination", which is a good description of what was actually measured. This should be used throughout the manuscript instead of 'memory', which is an interpretation (better brought up in the Discussion).

The text was edited accordingly (Lines 1, 35, 71, 75, 116, 164, 197, 209, 229, 343).

2. Methods: please expand on the methods used to count Fos+ nuclei (software for counting or manually counted? If manually, was the experimenter blinded to treatment?)

The method section used to count c-Fos was expanded as requested. The counting was done manually, and the experimenter was not blinded to treatment. A second experimenter who was blinded to treatment repeated the c-Fos-positive cell counting and the same results were obtained (Lines 657-659, 662).

3. Methods: please include a statement justifying use of male mice only.

A sentence in the method section was added to justify the use of male mice: “to reduce the variability due to female estrus cycle we restricted the analysis to male animals” (Line 419).

4. Results heading (line 72): "ACh released from cholinergic neurons in the MSDB is required for social novelty" should be '… social novelty discrimination'.

The text was modified accordingly (see point 1).

5. Legend in S2. (C and D) (and throughout) are not really a scatter plots.

“Scatter plots” were substituted with “aligned dot plots” accordingly.

6. Fos images are labeled with 'social behavior'. More accurate would be 'social interaction'.

The labels in c-Fos images were changed accordingly.

7. The axis labels for the Fos staining in Figure 3 suppl 2 (D) is unclear. In the methods, the authors state "for each biological sample, the percentage of CA2 Fos+ nuclei was calculated as a total from two hippocampal slices and normalized to that obtained in the HCC condition" but the axis is labeled "% of fos+ cholinergic neurons". Do the authors mean "% of cholinergic neurons that are Fos+"? Similarly for Figure 3 F "normalized % of Fos+ nuclei". Should be "% of nuclei that are Fos+ (normalized to HCC)".

In Figure 3—figure supplement 2D (New Figure 2D) we meant the % of cholinergic neurons that were cFos^+^. In the revised version of the manuscript we did not show normalized data to HCC in the CA2 region as well. We added a sentence in the method to explain that in MSDB, the % of c-Fos^+^ cholinergic nuclei was analyzed in HCC and social interaction conditions. The axis labeled in Figure 3 (F) (New Figure 4F) was changed to “% of cFos^+^ nuclei” as the Reviewer suggested.

8. The graphs in Supp 3-2 'D' and Figure 3F are low resolution.

The resolution of the graphs was increased (New Figure 2D and Figure 4F).

9. Please label CA1/CA2/CA3 on images.

CA1/CA2/CA3 labels were added to images.

10. Wording on lines 183-4: "In slices obtained from naive mice not expressing hM4, CNO did not affect the frequency and the amplitude of both sIPSCs (Figure 4-supplement 2C) and sEPSCs (Figure 4-supplement 2D)." should read: ".. CNO had no effect on either frequency or amplitude of sIPSCs or sEPSCs." Or something similar.

The sentence was changed accordingly: “In slices obtained from naïve mice not expressing hM4, CNO had no effect on either frequency or amplitude of sIPSCs or sEPSCs” (Lines 219-220).

11. Methods, line 494: spelling should be Xylazine, not Xilazine (unless brand name? please confirm).

The typing error was corrected.

12. Methods, line 432: please clarify or be consistent with usage of 'spontaneous' synaptic currents vs. 'miniature' currents. (stated: "Miniature GABAA-mediated inhibitory postsynaptic currents...", but abbreviate with sIPSCs)". If TTX was used in both cases, why the difference in nomenclature? (should be mEPSC and mIPSC in that case, no?).

The typing error was corrected.

13. Methods, line 498: "frost" should be "frozen".

The text was changed accordingly.

14. References: should cite Raam, et al., 2017, showing role for CA2 in social discrimination.

The reference was included in the Introduction (line 64) and in the Reference sections.

Reviewer #2:In this manuscript, the authors investigate the role of acetylcholine in social memory and sociability. They combine pharmacological experiments with chemogenetics, in-vivo experiments and different behavior paradigms. In general, the proposed question of how a common neuromodulator such as ACh regulate social memory is of general interest. The authors find that social novelty recognition it is mediated by ACh release from Medial Septum, which specifically activates nicotinic receptors in local interneurons, what contributes to disinhibition of CA2 pyramidal neurons. The main caveat of the study is the lack of temporal resolution for most of the points regarding the mechanisms by which ACh controls different aspects of social memory. The nature of these experiments assumes social memory as a whole part, while it has been shown that social (and in general any type of memory) it is further subdivided in different processes (encoding, consolidation, recall…) which in turn involve different brain mechanisms. Yet, I think that in general the authors made a good effort combining different techniques to dissect the present mechanisms that could potentially relate ACh with social memory. Some specific points that could strengthen the main conclusions are detailed below.1. In Supplemental Figure 1 (panel C), the authors show that after applying TeNT the frequency of spiking is not affected, however, there is a clear trend of a lower frequency with TeNT conditions, suggesting that there might be a masked effect due to low N.

We increased the number of MSDB ChAT^+^ neurons expressing eYFP (from 6 to 9, please note that one cell included in the previous version was excluded because it was an outlier, frequency = 14.9 Hz; ROUT method, Q=1%). Unfortunately we could not further increase the number of TeNT expressing neurons (n=8) because the viral injection of the AAV carrying TeNT did not lead to TeNTGFP expression. This was a consequence of the limited number of available animals due to the pandemic condition. In spite of the clear trend, the difference in the firing frequency did not reach the statistical significance by adding new eYFP cells.

2. In general, it could be nice to clarify in every part how long after infusion of different drugs were the animals tested, I couldn't find it for example for the TeNT experiments.

TeNT was durably expressed in ChAT-positive neurons in the MSDB starting at 3-4 weeks from viral infection. Both the behavioral and electrophysiological experiments were performed after 4 weeks from viral injection as described in the Methods section (Lines 442-443). Regarding the local drug infusions, behavioral experiments were performed 30 minutes after drug application (Lines 206, 234).

3. The cFos experiments are nice but lack the temporal resolution to correlate this activity dependent marker with any specifics of the task, i.e., social interactions/memory. It cannot be rule out that the cFos activity is free movement of the animal in any task (since there are no controls with any other behavioral paradigm). For this and in general for all group of experiments, it would be helpful to compare it with the responses obtained in the object recognition test (novel/familiar).

We agree with the Reviewer’s comment and, as requested also by Reviewers 1 and 3, we performed c-Fos analysis in animals exploring the three chamber arena in the absence of social stimuli (empty arena, EA). See point 4 of Reviewer 1.

4. In line 358 the authors claim that "local release of ACh in the CA2 is sufficient for social memory's encoding". I think that what it can be proved with their experiments is that local release of ACh in the CA2 is necessary for social memory's encoding. In addition, the title of this part states "Local release of ACh in the CA2 hippocampal region is necessary for social novelty", shouldn't it be "for social novelty encoding/detection/recognition"?

We agree with the Reviewer’s comment, the text (Lines 196-197) and the paragraph title were changed accordingly: "Local release of ACh in the CA2 is necessary for social memory encoding" and “Local release of ACh in the CA2 hippocampal region is necessary for social novelty discrimination”.

5. In the main text, it is stated that n=14/n=13 animals were used for optogenetics groups, while in the figure 4 legend it is n=12/n=10. Were some animals excluded maybe due to virus or canula mistargeting?

The Reviewer’s comment is correct: the animals treated with CNO locally (in CA2) were n=13/n=14 (eYFP and hM4 respectively), but some were excluded due to canula mistargeting (1 eYFP and 4 hM4). We then performed an additional set of experiments and increased the number of animals in both groups (eYFP: n=15; hM4: n=14) as indicated in the figure legend (Figure 5). The text was changed accordingly.

6. Could the authors provide any insight with immunostaining (or even just speculate/discuss it), on which type of interneurons might be being involved in disinhibiting CA2 cells?

Looking at different GABAergic interneuron subtypes would be very interesting but too demanding at this stage of the work. It warrants a completely new study. As mentioned in the discussion (Lines 395-398) “A disinhibitory effect of ACh was observed also in the prelimbic area of mPFC, where α5 nAChR dependent activation of VIP^+^ interneurons inhibits downstream SOM^+^ cells, which in turn leads to the enhancement of principal neurons firing in the layer 2/3 (Koukouli et al., 2017)”. Thus, VIP^+^ and SOM^+^ interneurons could be candidates for nicotinic-mediated disinhibition in the CA2 region. PV^+^ neurons could also play a role (Pi et al., 2013). Nevertheless, our double-immunostaining experiments for c-Fos and PV allow us to exclude a scenario in which PV^+^ interneurons are involved. These hypotheses will certainly be pursued of future investigations.

Reviewer #3:The current study by Pimpinella et al. investigates an important and timely question regarding the contribution of the neuromodulator acetylcholine to social memory. By conducting behavioral experiments with chemogenetic, pharmacological, and optogenetic manipulations of cholinergic neurons in the medial septum or cholinergic release sites in the hippocampal CA2 region in mice, the authors demonstrate that cholinergic signaling is critically important for social memory and also for social novelty-related neuronal activity in CA2. Cholinergic neuromodulation in the hippocampus has long been implicated in novelty encoding but the dependence of social memory on cholinergic modulation has not been shown so far. The authors further show that cholinergic modulation of social memory depends on nicotinic acetylcholine receptors as opposed to muscarinic receptors, an important step towards deciphering the mechanism of cholinergic action contributing to social memory encoding. The authors claim that they have identified the mechanism of cholinergic action underlying social memory formation as disinhibition of CA2 principal neurons via activation of nicotinic acetylcholine receptors on CA2 interneurons. However, this claim is currently only poorly supported by their experimental data and the authors do not rule out alternative hypotheses that may account for the observed effects. The authors further claim that the inhibition of cholinergic modulation in the CA2 region specifically affects social memory but no other types of memory such as spatial memory. However, they do not present convincing evidence that spatial memory is spared and that inhibition of cholinergic signaling is constrained to the CA2 region. In summary, the manuscript presents very interesting data on the importance of cholinergic signaling for social memory. However, the manuscript in its current form lacks critical evidence supporting the authors' claims regarding the proposed mechanism of cholinergic modulation of social memory.1. With respect to the TeNT experiments: The authors convincingly demonstrate that cholinergic MSDB neurons are important for social memory. However, they then further claim in lines 102-107 that the memory effect is specific for social memory based on comparing the effect on social memory with effects on novel object recognition. I am not convinced that this is the correct control experiment because social memory is hippocampus-dependent, while novel object recognition is not (or at least there is no strong evidence for novel object recognition being hippocampus-dependent). It is odd that the authors chose the NOR test as opposed to a hippocampus-dependent test such as the novel object location task. If the authors want to make the point that cholinergic signaling is specifically important for social memory, they would need to show that other forms of hippocampal-dependent memory such as spatial memory is unaffected or less affected by the same manipulations. Given the current data, it seems that hippocampal acetylcholine release from MSDB cholinergic projection neurons is important for hippocampus-dependent novelty tasks including-but not necessarily specific-to CA2-dependent social memory.

Although there is indication that the NOR task is hippocampus-dependent (see review Cohen and Stackman, 2015), the issue is controversial. Thus, we agree with the Reviewer’s comment concerning the need to show how cholinergic disruption impacts on spatial memory using a hippocampus-dependent test such as the object location task (OLT). To this end, ChAT-Cre mice injected with the AAV carrying either eYFP or hM4 sequences were subjected to OLT. CNO was systemically delivered via i.p. injection 30 minutes before mice were exposed to the new location of the object. Although hM4 mice showed preference to novel location as the eYFP group, the time they spent in exploring the object in the new location was not significantly different from that in the familiar location. This result suggests that inhibition of ChAT^+^ neurons in the MSDB affects hippocampal-dependent spatial memory. These data were included in a new supplementary figure (Figure 4—figure supplement 5).

2. Lines 89-91, Figure 2—figure supplement 1C: "No changes in the (…) spontaneous firing frequency between eYFP and TeNT expressing neurons were observed." The authors only compared n = 5 TeNT cells with n = 6 eYFP cells using a non-parametric test that is underpowered at a sample size of 5. Nevertheless, they observe a more than threefold reduction in firing rate with a p-value of p = 0.08. This is very close to significance despite the low sample size and the test being underpowered. It is very likely that the authors would find a significant difference if they increase the sample size to n > 10. I am concerned that TeNT does not only affect synaptic release of vesicles but also basic firing properties of the cells. The authors may argue that, even if that is the case, the main conclusion of the paper may not change because the overall effect of TeNT expression is an inhibition of cholinergic activity. However, it should be discussed that TeNT may not be specifically affecting only synaptic acetylcholine release.

The number of cells included in the analysis of TeNT group was 8 (circles shown in the figure overlap). We agree with the Reviewer’s comment on the possible effect of TeNT expression on neuronal firing activity, besides the one on acetylcholine release. We increased the sample of eYFP expressing neurons to 9 (see New Figure 3—figure supplement 1C). We could not increase the number of TeNT^+^ cells further (see answer to Reviewer 2-point 1). The additional data from eYFP group did not change the statistical outcome about the firing frequency (p=0.07, Mann-Whitney test). In addition, we performed a normality test (Shapiro-Wilk) in order to run a parametric test, of which the p value was 0.08.

We have two hypotheses regarding the TeNT-induced changes in intrinsic neuronal properties:

1. Diverse tropism of eYFP carrying AAV (serotype 2/9) *vs* TeNT carrying AAV (DJ serotype) may target different subtypes of cholinergic neurons in the MSDB.

2. TeNT could affect local (i.e. dendritic) ACh release with consequent effect on intrinsic properties modulated by cholinergic autoreceptors. This phenomenon was observed for neuromodulatory systems (i.e. dopaminergic neurons, see review article by Luwing et al., 2016).

3. With respect to systemic CNO experiments: It has been shown by Gomez et al. (2017) that CNO is very likely not the active component responsible for DREADD-effects but that CNO is instead converted to clozapine which then acts on DREADD receptors. The authors cite this paper and do the correct control experiments by comparing their results with CNO-injected eYFP mice. However, I am missing a more detailed discussion when introducing the method (line 120) and a short discussion about potential off-target effects that has been previously shown for the CNO concentrations used in the current study (10 µM for in vitro experiments and 100 µM or 3 mg/kg for in vivo experiments).

The discussion of the possible side-effects of CNO at the doses used in the current study was extended (Lines 150-152, 159-160, 202-204, 242-245).

4. Lines 145-147: "(…) social behavior elicited strong increase in c-Fos-positive cells (in the MSDB)". The authors do not provide evidence that c-Fos induction is caused specifically by social novelty as opposed to general novelty or exposure to the three-chamber test apparatus. The authors mention in the Discussion section that "(…) to rule out the contribution of c-Fos expression induced by a novel context, the animals were exposed to the three-chamber apparatus 24 hours before the social test" (lines 269-271). However, a single exposure to the test chamber one day before the test does very likely not reduce c-Fos expression on the next day. If the authors want to make that claim, they need to provide data showing that a single exposure to the same environment on the previous day significantly reduces c-Fos activation. As it stands, the current data do not show that social memory is related to c-Fos activation in the MSDB or hippocampus in addition to the c-Fos activation generally observed when taking mice out of their home cage and placing them in a test chamber.

We agree with the Reviewer that in the previous version of the manuscript we did not provide evidence that c-Fos activation in the MSDB and CA2 was caused by social novelty. As indicated in the answer to Reviewers 1 (point 4) and 2 (point 3), we performed c-Fos staining and quantification in the MSDB and in the CA2 hippocampal region of mice exposed to the three chamber empty arena (EA). We found a certain degree of activation in MSDB in animals exposed to EA (lower as compared to social stimuli). However, none of ChAT^+^ neurons analyzed were positive for c-Fos, indicating that ChAT^+^ neurons were selectively activated by social stimuli (New Figure 2). In the CA2 region of the hippocampus, we did not detect significant c-Fos activation in EA animals as compared to HCC group (Figure 4—figure supplement 3).

5. With respect to experiments using local injections of CNO or nicotine into the CA2 region, how can the authors be sure that CNO and nicotine (both injected at relatively high concentrations) do not diffuse to the surrounding areas CA1 and CA3? A caveat of those experiments is that the inhibition of cholinergic signaling may not be constrained to the CA2 region. Observed effects of cholinergic inhibition may therefore be completely or partially caused by blocking cholinergic action in surrounding areas CA1 and/or CA3. This point becomes even more important in the light of the results of the local CNO injection, which-contrary to what the authors argue in the manuscript-did not show a significant difference between eYFP and hM4 animals.Related to this point, the authors claim that "local release of ACh in the CA2 is sufficient for social memory's encoding" (line 168). However, this claim is not supported by their data. The data shown in Figure 4C clearly show that there is NO significant difference in the difference score between eYFP mice and hM4 mice (right panel). It would be wrong to draw that conclusion simply because there is a significant difference between Familiar and Novel in the eYFP group but not the hM4 group (see Nieuwenhuis et al., 2011). The correct statistical comparison is the one shown in the right panel (as acknowledged by the authors themselves earlier in the manuscript when introducing the difference score).

The nicotinic antagonists DHßE delivered to the CA2 area caused an impairment of social novelty. We agree with the Reviewer’s comment regarding the absence of statistical significance in the score between eYFP and hM4 groups treated with CNO locally. We performed an additional set of experiments delivering CNO in the CA2 region of eYFP- and hM4-expressing mice. We increased the number of animals to 15 and 14 for eYFP and hM4 groups, respectively. The difference score is now significantly different between eYFP and hM4 mice (New Figure 5).

Our attempt at collecting data on the diffusion of drugs infused *via* canula by using a lipofilic dye (DiI) was unsuccessful, probably because of the different diffusion properties of the dye as compared to the drugs used in the experiments. In our experiments we used a small volume (150 nL) and we always checked the canula placement, to be sure that all drugs targeted CA2. However, we cannot exclude some diffusion to neighboring CA3 and CA1 regions.

A similar consideration applies to data presented in Figure 4 —figure supplement 2. Since CNO can have off-target effects, the authors should statistically compare effects of CNO in hM4 mice to effects of CNO in naïve mice.

We performed additional experiments in naïve mice to increase the sample of cells recorded (IPSCs n=10 cells; EPSCs n=9 cells). To evaluate possible off-target effects of CNO, we compared the % of CNO-induced changes between naïve and hM4 groups for frequency and amplitude of sIPSCs and sEPSCs. These results were included in Figure 5—figure supplement 2.

6. Regarding experiments on nicotinic control of neuronal activity in CA2: The authors claim that the mechanism of nicotinic receptor-dependent social memory is disinhibition of principal neurons in area CA2 via nicotinic activation of interneuron-selective interneurons. While this is an intriguing model consistent with their data, the authors do not provide convincing evidence for such a model or mechanism. In particular, recordings of interneurons are missing to support the author's conclusion and alternative hypotheses have not been addressed.

We agree with the Reviewer that recordings from interneurons would greatly increase the understanding of the mechanism of nicotinic modulation. It is known that different types of interneurons are modulated by nicotinic receptors. To dissect out those involved in nicotinic mediated control of CA2 circuit would require a significant effort and imply the use of several mouse strains. Hence, we feel that this important endeavour is out of the scope of this work but will be surely be a matter for future investigations.

7. Previous studies (Malezieux et al., 2020 and Dannenberg et al., 2015) have shown that brain states associated with high cholinergic activity and theta oscillations result in reduced firing of CA3 principal neurons via activation of interneurons. If the net effect of acetylcholine release in CA2 is activation of principal neurons via disinhibition as opposed to inhibition of principal neurons as shown in CA3, that would be a very interesting finding.

As suggested by Reviewer 1, we obtained a new data set from slice recordings showing that nicotinic activation (both using nicotine or hM3-CNO strategy) decreases the frequency of spontaneous glutamatergic transmission in the CA3 region (Figure 7—figure supplement 2 and Figure 8). This suggests that indeed, in contrast to CA2, the CA3 region is inhibited by nicotinic activation. in vivo experiments using juxtacellular recordings combined with photo stimulation of ChAT^+^ neurons in the MSDB were performed to assess the effect of ACh release on the CA3 output. Photo stimulation did not change the frequency of spontaneous firing in CA3. We included these results in a new supplementary figure (Figure 9—figure supplement 2). This may be related to the concomitant activation of mAChRs, which would probably counteract the depressing effect of nAChRs on glutamatergic transmission.

However, the authors should discuss the effect of tiletamine/xylazine anesthesia in their experiments.

To address this point, in a subset of experiments we used a different mix of anesthetics (ketamine-xylazine). Photo stimulation of MSDB ChAT^+^ neurons under these conditions significantly increased the firing of CA2 bursting neurons, mimicking the results obtained in the presence of previously used anaesthetics (zoletyl-xylazine). These results were included in a new supplementary figure (Figure 9—figure supplement 1).

Moreover, the authors could address alternative explanations. For example, cholinergic stimulation could result in a general increase of network activity resulting in higher firing rates in both interneurons and principal neurons. Data on interneurons would help distinguish between those hypotheses.

Data from interneurons will be important, but we think it would be too demanding since we expect different interneuron behavior in response to photo stimulation of MSDB cholinergic inputs (i.e. different classes of interneurons should be identified by a post hoc morphological analysis).

8. I couldn't find details on optogenetics experiments in the Methods section (e.g., light power, wavelength, viral construct).

The information regarding the light power and wavelength was added in the Method section (Lines 599-603), the information pertaining to the viral construct that was used was already described (former line 329, now lines 425-432).

References:

The suggested references were added to the Reference list

Further comments:1. Title and elsewhere (e.g., line 69): "Septal cholinergic input (…) controls social memory (…)." The wording "controls social memory" is very vague. Can the authors describe the major finding of the manuscript more precisely?

The title was changed accordingly :“Septal cholinergic input to CA2 hippocampal region controls social novelty discrimination via nicotinic receptor-mediated disinhibition”.

2. Lines 29-30 and elsewhere: Patch clamp recordings in hippocampal slices are usually referred to as "in vitro" experiments as opposed to "ex vivo". Is there any specific reason why the authors have chosen "ex vivo" as opposed to "in vitro"?

We chose “ex vivo” since we think it is more appropriate for acute slices. “in vitro” definition is usually referred to cell culture or organotypic slices.

3. Lines 59: The authors could cite more recent reviews of the cholinergic system in addition to the reference to Teles-Grilo Ruivo and Mellor (2013), for example Dannenberg et al. (2017) and Haam and Yakel (2017).

The suggested Reviews were included in the manuscript (Lines 59-60).

4. Figure 2: I find the way the symbols and colors are used confusing. For example, the circle symbol is used for Animal, Object, Novel, and Familiar. It is further confusing that in E, orange and blue colors are used to indicate Novel and Familiar, but in B, D, and F, orange and blue colors are used to indicate eYFP and TeNT.

The colours of the objects in the schematic drawings were changed.

5. Line 173: "isolated pharmacologically". How?

The sentence “pharmacologically isolated spontaneous inhibitory and excitatory postsynaptic currents, were recorded before and after CNO (10 M) application in the presence of CNQX (10 µM), gabazine (10 µM) and physostigmine (3 M) to block AMPA, GABA-A receptors and acetylcholinesterase respectively” was added (Lines 215218).

[Editors’ note: what follows is the authors’ response to the second round of review.]

Reviewer #1:This study is an important contribution to the field and is likely of interest to a broad readership of neuroscientists. The revised manuscript is much improved from the first submission and presents some exciting new data. Although the authors addressed most of the reviewer concerns, there are still a number of issues that should be addressed in the text before it is acceptable for publication. These, along with some minor issues, are listed below.1. The authors now include data from Novel Object Recognition and Object Location tests (Figure 3E, F), however they still did not show whether investigation of a Novel Object induces Fos in a way similar to that induced by the social stimulus. The data now included showing Fos in response to exploration of an empty arena and home cage controls are important additions, but the possibility that an increase in Fos staining in CA2 may not be selective for a social stimulus should be clearly acknowledged and discussed since a similar response to a novel object has not been ruled out. This is an important point given that place field remapping occurs with novel objects to a similar degree as that occurring with a social stimulus (Alexander, 2016).

We agree with the Reviewer that our experiments do not rule out the possibility that, in the presence of a new object, the same degree of c-Fos activation would occur in CA2. We have now discussed this issue in the Discussion section “However, we cannot state that the increase in c-Fos staining in CA2 is selective for social stimuli since a similar response to a novel object has not been assessed (Alexander et al., 2016)”. Line 358

2. The authors should address the locations of the cannulas, several of which look like they were in CA1. In this case, they should acknowledge in the discussion the limitations of the method.

We think that the limitation of the method is stated in the sentence “Hence this finding warrants further support that ideally would be based on another experimental approach” already present in the Discussion section. Line 364. We added in the method section the maximal distance from CA2 we considered as an acceptable correct placement (within 350 µm from the CA2 border to CA1). Line 635

3. It was interesting that most of the Fos+ neurons in CA2 were likely interneurons, and so it is somewhat surprising that the authors still do not show recordings from interneurons. The findings do not appear to be overstated ( "…provides insight into the mechanism" in the abstract), but some discussion stating the limitations of the study is warranted. These recordings would be critical for supporting the authors model of cholinergic regulation of CA2 output, and so the authors are encouraged to submit those recordings as an eLife "Research Advance".

We agree with the Reviewer about the need to record from interneurons and we added a sentence in the discussion “Recordings from GABAergic interneurons will be critical to identify how different subtypes contribute to social memory via nAChR-mediated disinhibition”. Line 409

4. The graphic is confusing in light of (1) the new data showing the increase in Fos in the interneurons, (2) no recordings from interneurons, and (3) there is some discrepancy with the observation of increased firing of CA2 pyramidal cells in response to opto-stimulation. I suggest deleting the graphic unless it can be revised to better reflect the findings.

We agree with the Reviewer and we removed the graphic (Figure 10) from the manuscript.

Reviewer #2:In the revised version of the manuscript, the authors have added new experimental support to their previous point, which substantially helped to strengthen the points made. Increasing the n numbers for several measures also helped to make sure the conclusions are solid. I am more convinced now about the mechanism that they proposed, stating that ACh, via nicotinic receptors, is necessary for encoding social novelty. However, although is interesting that they provide several possible models at the end, I still regret the lack of more specific data in relation to the type of interneurons. Overall, I think the authors made a decent job tackling most of the points raised in the previous review round and the information provided in this manuscript will be altogether of interest for the community.Figure 1– Units in figure 1 for axon density? Should it be "fluorescence density (AU, X20)"?

The axis label refers to the density of fluorescent axons counted in bins of 50x50 µm from images acquired with a 20x objective as described in the Method section. Lines 669-678

– 676 line reads: "stacks of 7 focal planes with a distance of 2?m" should it be "stacks of 7 focal planes with a distance of 2mm".

The unit has been corrected as µm. Line 671.

– No background subtraction was used I assume?

The Reviewer is correct, no background subtraction was used.

Figure 2– This figure, still compares "stimulus" versus "non-stimulus" condition, without assessing whether the responses cFos+ are exclusive for an animal (social) or could be anything, object, spatial trajectory, foraging…

Experiments shown in figure 2 were performed to exclude c-Fos activation due to the new context, as discussed in point 1 raised by Reviewer 1.

Figure 7– Scales missing in A and C traces (or are they all the same one?), please clarify.

Scales in A and C are the same of E.

Figure 8– Scales missing in A and C traces.

Scales in A and C are the same of E.

Figure 9– Duration of the bursts would also be informative.

The duration of bursts was analyzed and no significant differences between control and light were detected (Control: 26.4 ± 6.9, Light: 25 ± 5.5 ms; p=0.62, Wilcoxon test).

– Number of bursts would also be informative.

The burst number was analyzed and no significant differences between control and light were detected (Control: 24 ± 4.7, Light: 29 ± 6.3; p=0.41, Wilcoxon test).

Figure 5—figure supplement 2– Temporal scale missing in traces showed in A.

Temporal scale in A and C is the same.

– Scales missing in traces showed in E.

Scales in E and G are the same.

– Y labels in I and J are confusing (shown in "%"), sIPSCs frequency is shown in Hz (in the above plots), so perhaps the authors meant difference between conditions or ratio compare to control in these plots?

In I and J we plotted the % of changes between pre- and post- CNO application to compare the difference among conditions (i.e. CNO effect in naïve vs hM4 expressing mice) as requested by Reviewer 3 (second question point 5) in the previous round of revision.

Figure 7—figure supplement 1– Temporal scale missing in trace showed in A.

Temporal scale in A and C is the same.

Figure 7—figure supplement 2– Temporal scale missing in trace showed in A.

Temporal scale in A and C is the same.

Figure 9—figure supplement 1– Temporal scale missing in trace showed in B.

Temporal scale in B and F is the same.

Figure 9—figure supplement 2– It is interesting that the spiking frequency of CA3 neurons is not affected at all by optical stimulation of MSDB, despite the frequency of sEPSCs being affected by nicotine.

We agree with Reviewer 2 that this is an interesting point. Our results may suggest that sEPSCs modulated by nicotine could be subthreshold events.

– Temporal scale missing in trace showed in B.

Temporal scale in B and F is the same.

Reviewer #3:The revised manuscript entitled "Septal cholinergic input to CA2 hippocampal region controls social novelty discrimination via nicotinic receptor-mediated disinhibition" by Pimpinella et al. now presents additional experimental data and analyses that address previous concerns raised by reviewers. In particular, the authors now demonstrate that local delivery of CNO to the CA2 region of hM4-expressing mice impairs social novelty detection. Furthermore, the authors added in vitro experimental data that demonstrate an increase in spontaneous EPSCs in the presence of nicotine in CA2 but not in CA1 and CA3. In fact, a decrease in spontaneous EPSCs is observed in CA3. These new data provide evidence that nicotinic cholinergic receptors differentially affect network activity in hippocampal subregions and support the authors' hypothesis that cholinergic disinhibition of CA2 via nicotinic receptors is important for social novelty discrimination. Furthermore, the authors now report experimental results that demonstrate that social interaction increases cFos expression in cholinergic neurons in the medial septal complex further highlighting a significant activation and contribution of the septal cholinergic system during social interaction. I have few remaining comments, mostly addressing statistical analyses and the clarity of data presentation.1. Regarding major point 2 (TeNT experiments): Do the authors now interpret their findings as an effect of TeNT on intrinsic firing properties? To more clearly distinguish between the alternative and null hypotheses that TeNT has or has not an effect on intrinsic firing properties, I suggest a statistical analysis using Bayes factor (Keysers, Christian, Valeria Gazzola, and Eric-Jan Wagenmakers. "Using Bayes Factor Hypothesis Testing in Neuroscience to Establish Evidence of Absence." Nature Neuroscience 23, no. 7 (July 2020): 788-99. https://doi.org/10.1038/s41593-020-0660-4.; Dienes, Zoltan. "Using Bayes to Get the Most out of Non-Significant Results." Frontiers in Psychology 5 (2014): 781. https://doi.org/10.3389/fpsyg.2014.00781). The Bayes factor will indicate whether (a) the data support the null hypothesis or (b) the data support the alternative hypothesis, or (c) the sample size is too low to draw a conclusion.

We performed a statistical analysis using Bayes factor with the aim to distinguish between the alternative and null hypotheses. The Bayes factor was calculated from http://pcl.missouri.edu/bftwo-sample. The Value obtained was 1.2, hence based on Keysers et al., 2020 we cannot draw a conclusion in favour of null or alternative hypothesis meaning that we cannot claim that TeNT had effect on intrinsic firing properties.

2. Why does the eYFP virus has a different serotype (2/9) than the TeNT carrying AAV (DJ serotype)? The possibility that diverse tropism of eYFP carrying AAV vs TeNT carrying AAV may target different subtypes of cholinergic neurons – as stated by the authors in the rebuttal letter – should also be discussed in the manuscript itself.

An AAV with DJ serotype carrying the eYFP reporter was not available at the time we performed the experiments.

The possibility that diverse tropism of eYFP carrying AAV vs TeNT carrying AAV may target different subtypes of cholinergic neurons – as stated by the authors in the rebuttal letter – should also be discussed in the manuscript itself.

Based on the statistical analysis suggested by the Reviewer we cannot draw any conclusion regarding the effect of TeNT on firing properties of MSDB cholinergic neurons. Thus, we think that discussing the issue of viral tropism is not relevant for the study. The properties of the viruses used in this study have been described in the Method section. Lines 426-434

3. To Figure 3C and Figure 4C: It is not immediately clear from looking at the figure if Novel and Familiar refers to an object or to an animal. I suggest to replace the circles with symbols that make this immediately clear to the reader (e.g., a drawing of a mouse in Figure 3C and drawing of an object in Figure 3A).

The circles have been replaced with drawings of mice in the main Figure 3 and 4.

4. Figure 4 —figure supplement 5: In my opinion, this figure should be part of main Figure 4 because it is an important control experiment and shows that chemogenetic inactivation of cholinergic neurons in the medial septal complex also affects spatial memory in addition to social memory, though the effect size appears smaller for effects on spatial memory compared to the larger effects on social memory. But I leave the final decision to the authors.

We decided to keep the figure concerning the OLT as supplementary because we think that adding the OLT experiments in the main Figure 4 would make it too crowded.

5. Line 211: "necessary" instead of "sufficient"?

The text has been changed accordingly. Line 209